



# Multi-Channel and Multi-Polarization Radar Measurements around the NEEM Site

Jilu Li[1], Jose A. Vélez González[1], Carl Leuschen[1], Ayyangar Harish[2], Prasad Gogineni[3], Maurine Montagnat[4], Ilka Weikusat[5], Fernando Rodriguez-Morales[1], John Paden[1]

[1]Center for Remote Sensing of Ice Sheets, University of Kansas, Lawrence, Kansas 66045, USA
[2]Department of Electrical Engineering, Indian Institute of Technology Kanpur, Uttar Pradesh 208016, India
[3]Department of Electrical and Computer Engineering, University of Alabama, Tuscaloosa, Alabama 35487, USA
[4]IGE, Université Grenoble Alpes, CNRS, IRD, G-INP, 38041 Grenoble, France
[5]Alfred Wegener Institute Helmholtz Centre for Polar and Marine Research, Am Alten Hafen 24, 27568 Bremerhaven, Germany

*Correspondence to*: Jilu Li (jiluli@ku.edu)

**Abstract.** Ice properties inferred from multi-polarization measurements, such as birefringence and crystal orientation fabric (COF), can provide insight into ice strain, viscosity and ice flow. In 2008, the Center for Remote Sensing of Ice Sheets (CReSIS) used a ground-based VHF radar to take multi-channel and multi-polarization measurements around the NEEM (North Greenland Eemian Ice Drilling) site. The system operated with 30 MHz bandwidth at a center frequency of 150MHz. This paper describes the radar system, antenna configurations, data collection, and processing and analysis of this data set. In the area of 100 km$^2$ around the ice core site, ice birefringence dominates the power variation patterns of co-polarization and cross-polarization measurements. The phase shift between the ordinary and extraordinary waves increases nonlinearly with depth. The ice optic axis lies in planes that are close to the vertical plane and perpendicular or parallel to the ice divide depending on depth. The ice optic axis has an average tilt angle of about 11.6° vertically, and its plane may rotate either clockwise or counterclockwise by about 10° across the 100-km$^2$ area, and at a specific location the plane may rotate slightly counterclockwise as depth increases. Comparisons between the radar observations, simulations, and ice core fabric data are in very good agreement.



## 1 Introduction

Radio echo sounding of ice sheets is a decades-old technique [Robin et al., 1969]. Ice thickness profiles, internal layers, bed topography and basal conditions revealed by radar echo sounding are widely used for analysis and modelling of ice mass balance, temperature, flow, and dynamics [Shepherd et al., 2012; MacGregor, 2015; Bamber et al., 2013; Hughes, et al., 2016]. Many previous studies show that ice properties, such as the birefringence and crystal orientation fabric (COF), could be inferred from multi-polarization radar measurements. The strength and polarization of radar echoes are affected by ice COF with respect to antenna orientations [Campbell and Orange, 1974; Bentley, 1975]. Ice COF is closely related to ice flow history and has a strong effect on ice deformation rate [Azuma and Higashi, 1985]; it is one of three reasons for radio echo reflections from internal ice layers [Fujita and Mae, 1994; Fujita, et al., 1999; Eisen, et al., 2007]. Hargreaves [1977] first proposed the theory of polarization and propagation of electromagnetic waves in a birefringent medium. He developed a method for determining radar wave polarizations by rotating a pair of orthogonal receiving antennas and observing variations in echo strength, and presented a linear incremental trend in the phase difference between ordinary and extraordinary waves caused by the ice birefringence from field measurements in the vicinity of DYE-3 in Greenland. Thereafter, many researchers further investigated this subject [Woodruff and Doake, 1979; Doake et al. 1981, 2002; Fujita and Mae, 1994; Fujita et al. 1999, 2006; Matsuoka et al. 2003, 2009, 2012; Drews et al. 2012]. Woodruff and Doake suggested that large changes in the birefringence of ice sheets at the grounding zone were caused by the effect of tidal strain on crystal orientation. This technique could distinguish the floating from the grounded areas of the ice sheets. Fujita and Mae proposed a dual frequency technique to distinguish permittivity-based and conductivity-based reflections from internal ice layers. Matsuoka et al. [2003] made multi-polarization radar measurements at two frequencies (60 and 179 MHz) from Dome Fuji toward the coast in East Antarctica, inferring the relationship between the identified COF spatial patterns of anisotropic reflectivity and birefringence to ice flow dynamics. Drews et al. [2012] tested and discussed in detail birefringence and other types of anisotropy in natural ice (e.g.





anisotropic distribution of elongated air inclusions) for their effect on radar data by applying different scattering models.

The NEEM project (North Greenland Eemian Ice Drilling), led by Denmark, was an international collaboration of 14 nations between 2007 and 2011 with the purpose of retrieving an ice core from the previous Eemian interglacial to understand the dynamics of past climates under conditions similar to the warming climate we are approaching. In the past, this ice core data has been used to infer the ice sheet's response to the strong warming in the early Eemian [NEEM community members, 2013] and derive ice and gas chronology [Rasmussen et al., 2013]. The fabric measured from the NEEM ice core and comparisons with GRIP (Greenland Ice Core Project) and NGRIP (North Greenland Ice Core Project) ice cores may help to constrain ice flow modelling along the ridge from GRIP to NEEM and onwards to Camp Century [Weikusat and Kipfstuhl, 2010; Eichler et al., 2013; Montagnat, et al., 2014]. In 2007, CReSIS (the Center for Remote Sensing of Ice Sheets) conducted both airborne and ground-based radar surveys to help identify the best area for the NEEM site. In 2008, CReSIS used an improved ground-based VHF radar capable of mapping the deepest layers in surveys along gridded lines covering about 100 $km^2$ around the NEEM site [Blake, et al., 2009]. Multi-channel and polarimetric measurements were collected with special antenna configurations along circular paths to infer changes in the ice fabric from radio echo sounding. The polarization measurements of this data set were preliminarily analysed in 2015 [Vélez González, 2015]. In this paper, we present further analysis to show that the ice birefringence dominates power variations in the received echoes from internal layers as the antennas rotate along the circular paths. We find that the phase shift between the ordinary and extraordinary waves increases nonlinearly with depth because of the ice birefringence. We infer the ice optic axis at the NEEM site is close to the vertical planes, approximately perpendicular or parallel to the ice divide for different depth ranges, and may rotate within a range between $-10°$(clockwise) and $+10°$ (counterclockwise) across the survey area. At a specific location, the plane also rotate slightly counterclockwise as the depth increases.

This paper first describes the surface-based radar system and antenna configurations used in the data collection of multi-polarization measurements, and gives details of the data processing techniques and analysis. Next, it compares the measurements with simulation results according to the theory of plane



wave propagation in a birefringent medium. Lastly, this study provides a comparison between radio echo sounding measurements and ice core observations with discussions and conclusions on the ice COF and birefringence properties around the NEEM ice core site.

## 2 Multi-polarization measurements

### 2.1 Radar system overview and antenna configurations

The radar system used in the survey was a modified ground-based version of the multi-channel radar depth sounder (MCRDS) developed by CReSIS in 2006 and 2007 [Lohoefener, 2006; Marathe 2008]. The system is composed of a digital section, RF (Radio Frequency) section, and antennas as shown in Fig. 1. The digital system is comprised of a waveform generator and a multi-channel data acquisition system. The RF section consists of two transmitters (denoted TX1 and TX2) and eight separate analog receiver channels with variable gain (denoted RX1 to RX8). Multiple receivers are used to provide beamforming capabilities in post processing. The digital waveform generator produces an offset video chirp from 15 to 45 MHz, which is up-converted to 135-165 MHz using a 120 MHz local oscillator mixing stage. The signal from the transmitter splits eight ways and feeds into four sets of two 50-Watt power amplifier banks (eight amplifiers for a combined peak power of 400 Watts). Each analog receiver is composed of a low-noise amplifier (LNA), a digitally controlled variable gain stage and an anti-aliasing filter with a 3-dB bandwidth of 130-170 MHz. The output signals from the analog receivers are captured by the data acquisition system using 12-bit analog-to-digital converters (ADCs) operating at a rate of 120 MSPS. Data from each of the receivers are stored in binary format with GPS-based time stamps for geo-location in post-processing. Table 1 summarizes the key parameters of the radar system.

The radiating structure consists of an array of twelve log-periodic antennas, each with a gain of about 6.7 dB. Each antenna is composed of 17 elements (dipoles). Fig. 2 shows the geometry and the radiation pattern of one of the 17-element log-periodic antennas [Harish, 2008]. Two antenna configurations were used during field operations: HH polarization setup for non-polarimetric measurements and quad-polarimetric setup for multi-polarization measurements (Fig. 3). In the HH polarization setup, both transmit and receive antennas were H polarized (i.e. the antenna plane is in the along-track direction).



The left and right transmitters (TX1 and TX2) were in front of the eight evenly spaced receivers (RX1~RX8). Each transmitter consisted of a pair of log-periodic antennas fed simultaneously by the RF transmitter to form a common phase center. In the quad-polarimetric setup, the antenna of TX2 and RX5~RX8 were V polarized by rotating them 90 degrees about the vertical axis (Fig.3, right). We

installed the antennas on a custom-made sled towed by a snow tractor (Fig.4, left). We also installed a GPS receiver between the transmitters and receivers to record geolocation.

## 2.2 Data collection

We retrieved multi-channel, multi-waveform, and multi-polarization data with the quad-polarimetric antenna setup on August 16, 2008.  Fig. 5a illustrates data collection paths. The survey paths began by

the NEEM ice core camp and went southwest perpendicular to the ice divide, passed through turning points 1, 2, 3 and 4 in sequence and returned to the starting point. The section from point 2 to 3 is along the ice divide at an angle of -60.7 degrees from north. The distances between locations 1 and 4 and between locations 2 and 3 are about 10 km. At each turning point, we travelled in a circular path with a radius of about 30 m to take quad-polarimetric measurements at all orientations (Fig. 5b). The circular

path at location 2 was not closed as at the other three locations.

During the data collection, the two transmitters TX1 and TX2 alternatively transmitted four waveforms (3-μs pulse by TX2, 3-μs pulse by TX1, 10-μs pulse by TX2 and 10-μs pulse by TX1) in sequence at a PRF (pulse repetition frequency) of 10 kHz. The signals from the eight receive channels were digitized separately. 64 consecutive pulses were stacked together and saved as one record. We thus took multi-

channel measurements of four polarizations (HH, HV, VH and VV) simultaneously. The speed of the snow tractor was about 3m/s, resulting in a distance of about 0.077 m between two data records.

## 3 Data processing and analysis



## 3.1 Receive channel equalization

While most polarization measurements previously reported in the literature were single channel data, we collected multi-channel polarization measurements for our purposes in this research. Although we can choose data from any one of the channels for analysis, we perform our analysis using combined multi-channel data because it has higher SNR (signal-to-noise ratio), reduced clutter and speckle artefacts, and clearly revealed deep ice layers. For a radar system with multiple receive channels and antenna arrays, any difference in hardware (e.g. cable length, amplifiers, attenuators and antenna matching/feed network) may result in channel mismatches in two-way propagation time delay, amplitude, and phase. Channel equalization — the process to compensate these channel mismatches — is required to combine the multichannel data to increase the signal-to-noise ratio and reduce the across-track clutter. We used co-polarization reflections from three distinct, specular, and continuous layers at depths of 1142m, 1181m, and 1271m (see Fig. 6a) to determine channel-to-channel differences. A total number of 12500 data records were used in this calibration step over a distance of about 1000 m along a straight path from the NEEM ice core site to Circle 1. Every 10 compressed pulses were stacked together and averaged (coherent integration) for better detection of these three layers; the two-way propagation time, amplitude, and phase were then extracted for each layer and channel, with 1250 samples. Channel RX2 and RX6 were selected as the reference channels for RX1~RX4 (H-polarization) and RX5~RX8 (V-polarization) respectively.

The two-way propagation time delay difference for each channel is:

$$\nabla t_d = \sum_{l=1}^{L}\{\sum_{m=1}^{M}[t_d(l,m) - t_{d,ref}(l,m)]/M\}/L \qquad (1)$$

where $l$ and $m$ are the sample and layer index respectively, $L = 3$ and $M = 1250$ are the number of layers and samples, and $t_d(l,m)$ and $t_{d,ref}(l,m)$ are the two-way propagation time delay for layer $l$ and sample $\boldsymbol{m}$ for each channel and the reference channel, respectively. The time delay calibration needs a higher order of accuracy compared to the interval of sampling. We determined the time delay differences by oversampling the data by a factor of 100 and performing cross-correlations with the reference channel.



To compensate for the time delay difference, we multiplied the data of each channel in frequency domain by $e^{j2\pi f \Delta t_d}$ where $f$ is frequency. We then determined the amplitude and phase mismatches $\Delta A$ and $\Delta \emptyset$ by comparing the complex values of peak reflections from the three ice layers. Table 2 lists the channel mismatches determined through this procedure. The time delay difference between channels is less than 8.2 nanoseconds, the amplitude mismatch between channels of the same polarization is less than 3 dB, and the maximum absolute phase imbalance is less than $180^{\circ}$.

Compared to single-channel measurements, the combined multi-channel measurements with channel equalization have more distinct internal layers, less speckle, and higher SNR. As mentioned earlier, these attributes are especially critical for the analysis of deep layers.

## 3.2 Transmit power mismatch

The power transmitted towards nadir by TX2 was about 12.7dB less than TX1. This transmit power mismatch is shown in Fig. 6, which includes averaged power-depth profiles (incoherent integration) along the straight line perpendicular to the ice divide. These mean power-depth profiles were generated from a total of 7248 data records over a distance of 523.19m. Here we compare four pairs of transmit-receive profiles: TX1-RX4 (HH), TX1-RX5 (HV), TX2-RX4 (VH) and TX2-Rx5 (VV). Fig. 6a, 6b, 6c and 6d show the power-depth profile differences between the four different polarization combinations. The embedded zoomed-in power profiles show the power differences between the reflected specular returns from layers at depths of 1050-1350 m, including the three distinct layers at depths of 1142, 1181 and 1271 m. The peak at 2520 m is the ice-bed interface.

The transmit power mismatch between TX1 and TX2 can be determined by looking at the power differences at 1142 m. As listed in Table 3, the echoed power levels of the four polarizations $P_{HH}$, $P_{HV}$, $P_{VH}$ and $P_{VV}$ from this layer are -107, -115.5, -123.4 and -121.2 dB, respectively. By defining the power level differences between the four polarizations as $\Delta P_a = P_{HH} - P_{VH}$, $\Delta P_b = P_{HV} - P_{VV}$, $\Delta P_c = P_{HH} - P_{HV}$, $\Delta P_d = P_{VH} - P_{VV}$, we thus have $\Delta P_a$, $\Delta P_b$, $\Delta P_c$ and $\Delta P_d$ equal to $16.4, 5.7, 8.5$ and $-2.2$ dB respectively.



$\Delta P_a$ contains the transmit power mismatch $\Delta P_{TX} = P_{TX1} - P_{TX2}$, anisotropic reflection power difference $\Delta P_{ANISO}$, and transmit-receive polarization mismatch $\Delta P_{VH}$ (see discussions in Section 3.3 about the polarization mismatch):

$$\Delta P_a = \Delta P_{TX} + \Delta P_{ANISO} - \Delta P_{VH} \tag{2}$$

$\Delta P_b$ contains the transmit power mismatch $\Delta P_{TX}$, $\Delta P_{ANISO}$ and transmit-receive polarization mismatch $\Delta P_{HV}$:

$$\Delta P_b = \Delta P_{TX} + \Delta P_{ANISO} + \Delta P_{HV} \tag{3}$$

$\Delta P_c$ contains the channel mismatch $\Delta P_{RX} = P_{RX4} - P_{RX5}$ and $\Delta P_{HV}$:

$$\Delta P_c = \Delta P_{RX} - \Delta P_{HV} \tag{4}$$

$\Delta P_d$ contains $\Delta P_{RX}$ and $\Delta P_{VH}$:

$$\Delta P_d = \Delta P_{RX} + \Delta P_{VH} \tag{5}$$

It is found that $\Delta P_{RX} = 1.5\ dB$ from the noise floor differences in Fig. 6c and 6d. The noise floor differences can include other effects (such as differences in the noise figure or electromagnetic interference) but the match between the following two estimates of $\Delta P_{TX}$ indicates that these differences

are small. Therefore from Eq. (5) we have $\Delta P_{VH} = \Delta P_d - \Delta P_{RX} = -2.2 - 1.5 = -3.7\ (dB)$, and it is derived from Eq. (2) that

$$\Delta P_{TX} = \Delta P_a + \Delta P_{VH} - \Delta P_{ANISO} = 16.4 - 3.7 - \Delta P_{ANISO} = 12.7 - \Delta P_{ANISO}\ (dB).$$

Similarly, from Eq. (4) we have $\Delta P_{HV} = \Delta P_{RX} - \Delta P_c = 1.5 - 8.5 = -7\ (dB)$, and it is derived from Eq. (3) that

$$\Delta P_{Tx} = \Delta P_b - \Delta P_{HV} - \Delta P_{ANISO} = 5.7 + 7 - \Delta P_{ANISO} = 12.7 - \Delta P_{ANISO}\ (dB).$$





The two $\Delta P_{Tx}$ estimates from cross comparisons match exactly. If the layer echo was from the isotropic zone (see ice anisotropy analysis in Section 3.3), then $\Delta P_{ANISO} \approx 0$ and $\Delta P_{Tx} \approx 12.7\ dB$.

We further confirmed that $\Delta P_{ANISO} \approx 0$ and $\Delta P_{Tx} \approx 12.7\ dB$ by comparing the peak power levels of HH (TX1-RX4) and VV (TX2-RX5) measurements along Circle 3, at depths in both isotropic and anisotropic zones. Along a small circle, we expect the power level of the peaks of the HH and VV measurements to be the same for two locations $90°$ apart on the circle because the antenna azimuth orientations of the H and V configurations are the same. However, the average power level difference between HH and VV measurements for peak pairs ~$90°$ apart on the circle (pk1-pk3, pk2-pk4, pk5-pk10, pk6-pk11, pk11-pk12, pk8-pk9) is ~12.7 dB (see fig. 7). The power variation patterns in Fig. 7 were obtained after the data were filtered by the 2-D moving filter described by Eq. (13) and (14) and discussed in Section 3.3.

The 12.7-dB transmit power mismatch between TX1 and TX2 is because of the truss effect and the mutual coupling between the two antennas of the transmitter in TX2, resulting in the main lobe of the radiation pattern pointing to an off-nadir direction, and other reasons currently unknown. We performed full-wave electromagnetic simulations to analyse the effect of the truss on antenna radiation characteristics, confirming the truss effect. Simulation results have shown that the radiation pattern is similar to Fig. 2b if the antenna is perpendicular to the truss, and tilts about $36°$ (Fig. 8) if the antenna is parallel to the truss, resulting in ~3dB transmitted power reduction [Harish, 2008]. The real truss structure is much more complicated than the simulated one, and it may result in larger tilt and power reduction. In addition, the unconsidered mutual coupling may form nulls in the radiation pattern. According to Fig. 6a, the SNR of echoes from deep layers at depths of 1753, 1832, and 1894 m are 8.5, 11.5, and 9 dB, respectively, for TX1-RX4. However, these layers are barely visible for TX2-RX4 because of the reduced transmit power and power loss from the transmit-receive polarization mismatch. It may be worth investigating if the TX2 transmit issue could be avoided or alleviated by rotating TX2 in Fig. 3a ninety degrees instead of the arrangement in Fig. 3b. Although the transmit power towards nadir by TX2 was reduced, the system had adequate performance for polarimetric measurements because of the redundancy of the quad-polarimetric configuration. Hereinafter, we only analyze polarization measurements from transmit TX1, which includes HH and HV transmit-receive polarizations.





### 3.3 Circular data analysis

Figure 9 presents the HH and HV polarization measurements at Circle 3. For HH measurements, the data from RX1, RX2, RX3 and RX4 were averaged together after compensating for inner-channel mismatches (as discussed in Section 3.1). The same procedure was used for HV measurements with data from RX5,

RX6, RX7 and RX8. To further increase the signal-to-noise ratio and to reduce speckle, two data records were coherently averaged along the circle, followed by four incoherent integrations. The coherent and incoherent integrations resulted in a sampling interval of about 0.54 m along the circle, corresponding to an azimuth change of about $1.125°$.

In Fig. 9, the top 250 m is blank. This corresponds to the eclipsing time, which is equal to the duration of

the 3-μs chirps that the radar has to wait before recording data for unambiguous range detection. The sudden intensity drop between the depth of 1450 m and the ice bed at 2500 m corresponds to the so-called echo free zone (EFZ). As seen in Fig. 9, the receive power level obtained for HH measurements is higher than that of HV measurements for shallow depths between 250 and 750 m, and the power level of HV measurements becomes comparable to that of HH measurements for depths of 750 to 1450 m.  Even in

the EFZ, we observe three distinct layers at depths of 1753, 1832 and 1894 m, respectively. The most unique and impressive feature in Fig. 9 is the obvious signal distinction patterns separated by about $90°$, in both the HH and HV measurements of middle depth range (between 750 and 1450 m).

Next, we will compare results from polarimetric measurements with simulations based on existing theory [Hargreaves, 1977] and show that the above patterns are caused by dominant ice birefringence effects.

We will treat the nadir-transmitted radar signals into the ice sheets as plane waves traveling downward. If we take the nadir as the positive z-direction, then the electric field of the plane wave $E$ is in the x-y plane perpendicular to the z-axis. Its components $E_x$ and $E_y$ are given as:

$$E_x = A_x \cos(\omega t - kz) \tag{6}$$

$$E_y = A_y \cos(\omega t - kz + \emptyset) \tag{7}$$





where $A_x$ and $A_y$ are the amplitude of the x-component and y-component respectively, $\omega$ is the angular frequency, k is the wave number and $\emptyset$ is the phase difference between the two components. **E** is a function of time at any fixed position z, and the vector tip will trace a curve in the x-y plane as time elapses. The plane wave is of linear, circular, or elliptical polarization when this curve is a straight line, a

circle, or an ellipse [Ulaby, 1981]. In general, Eqs. (6) and (7) describe an elliptical polarization. Circular polarization is a special case when $A_x = A_y \neq 0$ and $\emptyset = \pm 90^\circ$, and linear polarization is a special case when $\emptyset = 0$ and $A_x$ and $A_y$ are not zeros, or when either $A_x$ or $A_y$ is zero.

The usual dipole and Yagi antennas radiate linear polarized waves (in general, $\emptyset = 0$ and $A_x$ and $A_y$ are not zeros). The polarization and antenna planes match. Two identical and orthogonally-crossed dipoles

fed with a $90^\circ$ phase difference radiate circular polarized waves [Balanis, 2005]. The 17-element log-periodic antennas radiate linear polarized waves and the polarization planes match with the antenna planes. Ideally, if there is no polarization change of radio waves during propagation, there is no signal loss from polarization mismatch if the transmit (TX) and receive (RX) antennas are co-polarized (HH or VV); otherwise, there is a signal loss because of the polarization mismatch between the TX and RX

antennas, with cross-polarized TX-RX as the extreme case (HV or VH). Previous studies have found that the polarization state of radio waves might change after propagating through and reflected from ice sheets because of ice properties such as the birefringence of ice [Bogorodsky, 1976]. In these cases, the signal loss would occur from polarization mismatch — even if the TX and RX antennas are co-polarized.

Ice sheets are polycrystalline — formed by many ice crystals of different shapes and sizes at different

orientations. The orientation of an ice crystal can be determined by its c-axis orientation. The crystal orientation fabrics (COF) in ice sheets are classified based on the c-axis projection distribution on Schmidt diagrams. Vertical single-maximum, elongated single-maximum, and vertical girdle are three typical COF types usually found in ice cores from ice sheets, mainly drilled at divides and domes [Alley, 1988; Matsuoka et al., 2003; Fujita et. al. 2006; Eisen et al., 2007; Montagnat et al. 2012; Weikusat et al. 2017;

see Faria et al., 2014 for a review]. For single-maximum COF, all c-axes align up in a cone centred along or close to the z-axis, and in the classical glaciological projection into the horizontal the points in Schmidt diagrams (stereographic projects) concentrate close to the center of the circle as this center represents the

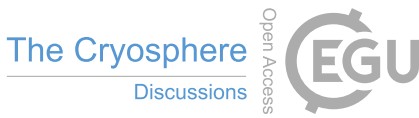

vertical direction ($z$ axis). The point distribution of elongated single-maximum COF is an ellipse around the center of the circular diagram. A band across the diagram is called vertical girdle COF, representing a great-circle in the un-projected reality. These three COF cases correspond to thinning dominated flow or radial dome-flow situations (vertical single maximum COF), diverging or parallel flow (girdle COF), and general flow situations with diverging and thinning components (elongated maximum COF) in ice sheets, as ice crystals rotate differently under these various compression and extension situations [Faria et al., 2014a; Paterson, 1994].

Different COF types result in different dielectric permittivity properties which can be described by the dielectric permittivity tensor $\epsilon_p$ in the principle coordinate system:

$$\epsilon_p = \begin{bmatrix} \epsilon_{x_p} & 0 & 0 \\ 0 & \epsilon_{y_p} & 0 \\ 0 & 0 & \epsilon_{z_p} \end{bmatrix} \tag{8}$$

where $z_p$ is parallel with the symmetric axis of the COF, $\epsilon_{x_p}$ and $\epsilon_{y_p}$ are the two components in the $x_p$-$y_p$ plane that is perpendicular to $z_p$. For single-maximum COF, $\epsilon_{x_p} \approx \epsilon_{y_p}$; for elongated single-maximum COF, $\left| \epsilon_{z_p} - \epsilon_{x_p} \right|$ or $\left| \epsilon_{z_p} - \epsilon_{y_p} \right| \gg \left| \epsilon_{x_p} - \epsilon_{y_p} \right| > 0$; and for vertical-girdle COF, $\left| \epsilon_{z_p} - \epsilon_{x_p} \right|$ or $\left| \epsilon_{z_p} - \epsilon_{y_p} \right| > \left| \epsilon_{x_p} - \epsilon_{y_p} \right| > 0$. For ice with elongated single-maximum COF, the dielectric permittivity is about the same around the $z_p$-axis and the difference between $\epsilon_{z_p}$ and $\epsilon_{x_p}$ or $\epsilon_{y_p}$ is about 0.034 according to previous study [Matsuoka et al., 1997]. This uniaxial symmetry and anisotropic property is known as the birefringence of ice sheets.

For convenience, if the ice is birefringent, we can choose the x-axis and y-axis so that the optic axis of the ice is in the y-z plane and along the principal axis $z_p$, which is at an angle of $\beta$ to the z-axis (see Fig. 19c). According to the solutions of Maxwell's equations, a plane wave transmitted vertically down the z-axis to a birefringent ice sheet would split into an ordinary wave (its electric field is in the direction of





the x-axis) and an extraordinary wave (its electric field is in the y-z plane). The wave numbers of the two waves $k_1$ and $k_2$ would be different, which results in a phase difference at depth z given by

$$\emptyset = \emptyset_2 - \emptyset_1 = 2(k_2 - k_1)z \tag{9}$$

where $k_1$ and $k_2$ are given by Eq. A4a and A4b in [Hargreaves, 1977]. Therefore, for $\beta \neq 0$, and the

transmit polarization not aligned with the x- or y-axis, the transmitted wave of linear polarization would become elliptically polarized after passing through the birefringent ice because of the above phase shift.

When denoting the angle between the TX1 plane and the x-axis as $\alpha$, the received reflections from ice layers for a receiver positioned at an angle of $\gamma$ with respect to the x-axis would be

$$E_x' = A_x' \cos\alpha \cos\gamma \cos(\omega t - kz) \tag{10}$$

$$E_y' = A_y' \sin\alpha \sin\gamma \cos(\omega t - kz + \emptyset) \tag{11}$$

where $A_x'$ and $A_y'$ are the amplitude of the $x$ and $y$ components respectively, which include the losses from propagation, ice attenuation and reflections. Therefore the power of the received signals is

$$P_z(\alpha, \theta, \emptyset) = E_x'^2 + E_y'^2 = A_x'^2 \cos^2\alpha \cos^2\gamma + A_y'^2 \sin^2\alpha \sin^2\gamma + \frac{1}{2}A_x' A_y' \sin 2\alpha \sin 2\gamma \cos\emptyset \tag{12}$$

For RX1~RX4, the antenna plane is parallel with that of TX1 (co-polarization), so $\gamma = \alpha$; for RX5~RX8,

the antenna plane is perpendicular to that of TX1 (cross-polarization), so $\gamma = \alpha + 90^0$. Fig. 10a presents the simulation results of the received signal power as a function of α and $\emptyset$ for the co-polarization and cross-polarization cases respectively based on Eq. (12) assuming $A_x' = A_y' = 1$. The patterns repeats every $90°$ of α and are therefore displayed only in the range of $0°{\sim}90°$ for α. Each pattern is symmetric with respect to $45°$ and$180°$ for α and $\emptyset$, respectively. The co-polarization power has minima at α = $45°$

and $\emptyset = 180°$, and the cross-polarization power's maxima are at the same locations.

The solid lines in Fig. 10b compare the simulated power level of the received signals of the co-polarization and cross-polarization for phase shifts of $30^0, 60^0, 90^0$ and $120^0$ (the phase shift increases with depth) as α rotates from $0°$ to $360^0$. The extinctions of cross-polarization power occur at $\alpha =$



$0^0, 90^0, 180^0, 270^0$ and $360^0$. At $\emptyset = 30^0$, the co-polarization power level varies little as α rotates. The minima at $\alpha = n90^0 + 45^0 (n = 0,1,2,3)$ are only 0.3dB below the maxima (0dB) at $\alpha = n90^0 (n = 0,1,2,3,4)$ ; the maxima of the cross-polarization power at $\alpha = n90^0 + 45^0 (n = 0,1,2,3)$ are 11.75dB down with respect to the maximum co-polarization power. At $\emptyset = 60^0$, the co-polarization power

variations become visible as α rotates. The minima are 1.25dB down from the maxima, and the maxima of the cross-polarization power increase to -6dB. At $\emptyset = 90^0$, the co-polarization power has minima of -3dB, and the maxima of cross-polarization power match the co-polarization minima. At $\emptyset = 120^0$, the co-polarization power has minima of -6dB and the maxima of the cross-polarization power increase to -1.25dB and become comparable with the maximum co-polarization power.

Previous surveys involving polarimetric measurements were performed along a short straight line by moving back and forth with fixed antenna orientation, and repeating the measurements after changing the antenna orientation [e.g. Fujita et al., 1999; Matsuoka et al., 2003]. This way, multiple measurements along a straight line for a fixed antenna orientation could be stacked together to reduce the fading of received echoes. In our case, we cannot directly stack measurements together along the circular paths

because the antenna orientations are continuously changing during the measurements. We therefore filtered the HH and HV measurements of Circle 3 with a windowed 2-D moving average filter, described by:

$$P'_{ij} = \sum_{m=-M/2}^{M/2} \sum_{n=-N/2}^{N/2} h_{mn} P_{i+m,j+n} \tag{13}$$

where $P'_{ij}$ is the filtered value of power, $i$ and $j$ are the range bin and along-track indexes of the data

matrix, $P_{mn}$ is the value of power before this filtering, $M$ and $N$ are the window length and $h_{mn}$ is the 2-D filter coefficient calculated by:

$$h_{mn} = \frac{w_{r=m+\frac{M}{2}+1} w_{s=n+\frac{N}{2}+1}}{\sum_{r=1}^{M+1} w_r \sum_{s=1}^{N+1} w_s} \tag{14}$$

in which $w_r$ and $w_s$ are the window weights in range bin and along-track dimensions. This filtering removes the adverse fading effects from our analysis.



Instead of considering the reflected power from a few specific and isolated layers at different depths (as in previous studies), we looked at the bulk properties of the received power at all depths. For every 10 range bins between 100 and 650 (corresponding to depth increments of about 28 m between depths of 276 and 1826 m), we calculated the co-polarization and cross-polarization power profiles against the

angle between TX1 and the north. Fig. 11 presents the HH and HV power profiles at depth increments of about 112 m from the top to the bottom of the plots. These profiles were obtained after filtering the data by Hanning windows with $M = 51$ and $N = 11$. Fig. 12a visualizes the filter kernel applied to convolve with the power intensity matrix. Fig. 12b compares the HH and HV power profiles at depth of 806 m before and after the filtering, illustrating that this operation can effectively smooth out the fading and

extract the power variation patterns.

By integrating Eq. (12) over $[0,2\pi]$ of $\alpha$ for co-polarization and cross-polarization cases, the birefringence phase shift as a function of ice depth $z$ can be solved as

$$\emptyset = cos^{-1}\left(\frac{1-3P_R}{1+P_R}\right) = cos^{-1}\left\{\left[1 - \frac{\frac{3}{2}\max(P_{z_{cr}})}{\bar{P}_{z_{co}}}\right]\middle/\left[1 + \frac{\frac{1}{2}\max(P_{z_{cr}})}{\bar{P}_{z_{co}}}\right]\right\} \tag{15}$$

where $P_R = \bar{P}_{z\_cr}/\bar{P}_{z\_co}$ is the ratio of the received average cross-polarization power ($\bar{P}_{z\_cr}$) to average

co-polarization power ($\bar{P}_{z\_co}$) at depth $z$. Based on Eq. (15), within a range of depth we could infer the birefringence phase shift using the obtained HH and HV power profiles at different depths.  We ignored the power profiles at depths shallower than 324 m because the filtering uses truncated data at depths less than 254 m for the 3-μs waveform. We also ignored the power profiles deeper than 925 m, in which the maximum cross-polarization power becomes larger than twice the average co-polarization power,

resulting in no real solutions from Eq. (15). The red curve in Fig. 13a presents the estimated birefringence phase shift at Circle 3 as a function of depth. The plot shows that the phase shift between the ordinary wave and extraordinary wave increases nonlinearly.  Between the depths of 333 and 755 m, the phase shift increases relatively slowly by $36°$ (from $50°$ to $86°$), and between the depths of 755 and 925 m, the phase shift increases by $71°$ (from $86°$ to $157°$ ) — almost 5 times faster. The red circles in the plot

correspond to phase shifts of $60°$, $90°$ and $120°$ at depths of 405, 700 and 806 m respectively. The HH




and HV power profiles (blue and red dots in Fig. 10b) at these three depths are overlapped over the simulated results (blue and red solid lines). For comparison, we converted the angle between TX1 and the north in Fig. 11 to $\alpha$ — the angle between Tx1 and the ice divide ($\alpha = 0°$ when the plane of TX1 is along the ice divide) by circularly shifting the curves to the right by $60.7°$ (the angle between the ice divide and

the north). We also shifted the level of the HH and HV power profiles up by 93.6, 103.3 and 111.7dB — the values of the maximum HH power below 0 dB at each depth. We observe that the variation patterns of both the HH and HV power profiles match well with the simulated ones. The HV power level is lower than the HH at $\emptyset = 60^0$, becoming comparable with HH power levels at $\emptyset = 90^0$ and $120^0$. The angles where the maxima and minima occur align with the simulated ones, although there are minor offsets. We

therefore conclude that the HH and HV power variations are mainly determined by the ice birefringence. We notice that the major mismatch between the measured power profiles and the simulated ones are with the HH, where its maxima at $\emptyset = 0^0, 180^0$ and $360^0$ are relatively smaller than the ones at $\emptyset = 90^0$ and $270^0$. This mismatch may come from the anisotropic nature of the reflections not taken into account in the simulation.

We performed similar analyses of the data obtained from measurements performed at the other three circles. Circle 2 was not completed during data collection; therefore we used the measurements from the half circle and completed the circle by repeating the measurement according to the power variation periodicity. The derived birefringence phase shift-depth profiles are also shown in Fig. 13a in blue, green and dark curves respectively for Circle 1, 2 and 4. The minima of the cross-polarization measurement

power profiles are the most distinct and unambiguous feature shown in Fig. 9b and Fig. 11b; therefore, we tracked the antenna orientation angles between TX1 and the north, where the minima occur in four circle locations at depths not greater than 1404 m (see Fig. 13b). We observe the following major characteristics from spatial variations in Fig. 13:

(1) The nonlinearities of birefringence phase shifts are similar for all four circles (e.g. plateaus between
depths of 700m and 800m, sudden and faster increases in the phase shift after the plateau).





(2) While the phase shift profiles of Circle 2, 3 and 4 are close to each other, Circle 1 is departed from the others; the phase shift of the plateau is at least 26 degrees higher. In Fig. 13a, we observe a faster phase shift slope before and after the plateau at Circle 1, which may suggest greater anisotropy at Circle 1 than other circle locations.

(3) The locations of the HV power minima at Circle 1 and 2 are very close to each other. Their offsets from the ones at Circle 3 are less than $10°$ (rotate clockwise) on average for most depths, and the minima at Circle 4 are constantly smaller than those at Circle 3 by around $10°$ (rotated counterclockwise) on average for all depths (see Fig. 19b).

(4) At all circles, we observe the general trend of the angle to the north of the HV minima rotating slightly
counterclockwise as the depth increases, which implies that the plane of the ice optic axis rotates slightly counterclockwise as the depth increases.

The three curves at the bottom of Fig. 11 correspond to depths of 1516, 1629 and 1742 m in the EFZ. The HV power variations still follow the birefringence patterns, and the rotation of the plane of the ice optic axis continues.

Based on Fig. 13b, we could only infer that the ice optical axis is in a vertical plane that is either parallel or perpendicular to the ice divide. However, we could remove this ambiguity by investigating the anisotropy in the power variation patterns of HH measurements. Fig. 11a marks three distinct power variation patterns of the HH measurements we identified and used to characterize the ice anisotropic properties: a) for ISO, the ice is isotropic around axis $z$ with no obvious power peaks at shallow ice depth
when birefringence phase shift is small or with four power peaks at about the same level; b) for ANISO-1, the ice is anisotropic between $x$ and $y$ axes, with two higher power peaks and two lower peaks interleaved; c) ANISO-2 is similar to ANISO-1 with an orientation shift of $90°$. Both ANISO-1 and ANISO-2 result from the effects of anisotropic reflections at internal ice layers. Because the reflection coefficient along the direction of the extraordinary wave field ($y$ axis) is larger than along the ordinary
wave field ($x$ axis) and the two higher power peaks are at orientations perpendicular to the ice divide in

ANISO-1, ANISO-1 indicates that the ice optic axis is in the $y$-$z$ plane perpendicular to the ice divide and ANISO-2 indicates that the ice optic axis is in the $y$-$z$ plane parallel to the ice divide.

Figure 14a presents the depth profiles of the above three power variation patterns traced from the HH polarimetric measurements at the locations of the four circles. The transition patterns between the boundaries of any two patterns are not considered and plotted in the charts. In Fig. 14a, we can observe the following characteristics:

(5) For shallow depth above ~450 m, no anisotropic property is observed between the x- and y-axes, and the ice is in ISO pattern.

(6) Between ~450 and 1065 m, at all circle locations the ice is anisotropic between the x- and y-axes, and the ice optic axis is in a plane that is approximately perpendicular to the ice divide. This anisotropic property continues to ~1319 m at Circle 2. Below the first ANISO-1, the ice returns to ISO pattern for depths from 1094 to 1178 m, from 1263 to 1319 m, from 1094 to 1319 m, and from 1065 to 1375 m at Circle 1 to Circle 4 respectively. There is a second ANISO-1 pattern at Circle 1 and Circle 2, between 1206 and 1347 m and between 1347 and 1375 m respectively.

(7) There is an ANISO-2 pattern at all circle locations and the optical axis is parallel to the ice divide; this occurs at depths from 1375 to 1545 m, from 1404 to 1545 m, from 1347 to 1407 m and from 1404 to 1545 m respectively at Circle 1 to Circle 4; the pattern switch from ANISO-1 to ANISO-2 at Circle 1 and Circle 2, from ISO to ANISO-2 at Circle 3 and 4.

(8) For depths below ANISO-2, no anisotropic property is observed between the x- and y-axes, and the ice is in ISO pattern.

We also observed the above isotropic and anisotropic patterns in the data collected with the non-polarimetric antenna configuration (see Fig. 3a) at crossovers along grid lines that are either parallel or perpendicular to the ice divide. Fig. 14b presents an example of stacked and normalized power-depth profiles at one of these crossovers, illustrating the isotropic and anisotropic patterns at different depth ranges. At this crossover, from top to bottom the ice column has ISO (324~450 m), ANISO-1 (450~1000



m), ISO (1000~1340 m), ANISO-2 (1340~1500 m) and ISO (1500~2000m) patterns. Fig. 14c displays the parallel (blue segment) and perpendicular (red segment) paths at the crossover. Each segment has 40 seconds of travel. The grid line data of these two segments were collected on the same day (Aug 12, 2008) without any radar setting changes. The data were SAR (synthetic aperture radar) processed first, and then the multiple power-depth profiles within about 120-m distance for each segment were averaged to obtain the stacked power-depth profiles which were normalized to the maximum power of the perpendicular segment. The cross mark in Fig. 5a shows the location of the crossover related to the four circles. Fig. 14d and Fig. 14e present the details of the power-depth profiles of the ANISO-1 and ANISO-2 patterns, respectively. For the power-depth profiles of ANISO-1 pattern, the power received along the segment perpendicular to the ice divide is constantly higher than that along the segment parallel to the ice divide, and the power difference is 1.5 dB on average, with a maximum difference of 6.3 dB at the internal ice layer interface. For the power-depth profiles of ANISO-2 pattern, the power received along the segment perpendicular to the ice divide is constantly lower than that along the segment parallel to the ice divide, and the power difference is 1.9dB on average, with a maximum difference of 4.4 dB at the internal ice layer interface.

## 4 Comparison with ice core measurements

The COF along the full NEEM ice core was measured in the field by an automatic ice texture analyzer every 10 m from a depth of 33 m down to 2461 m [Weikusat and Kipfstuhl, 2010], presented using the second-order orientation tensor $\boldsymbol{a}^{(2)}$ [Eichler et al., 2013; Montagnat, et al., 2014]. The eigenvalues of $\boldsymbol{a}^{(2)}$ represent the lengths of the ellipsoid axes best fitting the density distribution of the ice crystal c-axis orientations; its eigenvectors give the directions of the axes of the ellipsoid. Fig. 14 displays the eigenvalues of $\boldsymbol{a}^{(2)}$ side by side with the radar echogram of HV polarization measurements at Circle 3 adopted from Fig. 9b for a visual comparison (the radar echogram is detrended to reduce the dynamic range for clearly displaying the weak layers close to the ice bed). According to Fig. 14, the COF around NEEM can be characterized with five sections (A, B, C, D and E) along the whole ice column from surface to the bed. For section A (From ice surface down to ~ 400m), the COF is the isotropic single maximum ($a_3^{(2)} \approx a_2^{(2)} < 1/3$) and we cannot observe the HV power distinction pattern. The COF

evolves towards an elongated single maximum ($a_3^{(2)} < a_2^{(2)} < 1/3$) from shallow to deep depths in section B with an anisotropic fabric for depths down to ~1400 m, and the HV power distinction pattern becomes more obvious as the depth increases. The sudden drop both in COF eigenvalues and radar signal intensity in section C between depths of ~1400 to ~1500 m corresponds to the Wisconsin-Holocene

climate transition and results in an strengthened and sharpened single maximum COF ($a_3^{(2)} \approx a_2^{(2)} \ll 1/3$) in section D between ~1500 and ~2200 m. The HV power distinction pattern of birefringence continues from section B through C and D to the three distinct deep ice layers at 1753, 1832 and 1894 m. In section E from ~2200m to the bottom, the ice is stratigraphically folded. The COF is multi-clustered in certain layers. Beyond 1894 m, we observe weak ice layers close to the ice bed from radar

measurements.

The HH power variation patterns revealed by Fig. 14 match with the second-order orientation tensor $\boldsymbol{a}^{(2)}$ shown in Fig. 14a. Roughly the pattern is: ISO in Section A, ANISO-1 in Section B, ANISO-2 in Section C and ISO in Section D. The orientation of the crystals in the core is only known relative to the vertical axis. The orientation of the core in the x-y plane after it was extracted is not known. Therefore, direct

checks of ANISO-1 and ANISO-2 are not possible with the core data. In the following paragraphs, we compare the vertical orientation information revealed by radar polarization measurements and ice core data.

Figure 16 includes the pole figures (stereographic projections) projected into the sample plane (vertical) for depths at 330, 561, 849.8 and 1025.8 m, in which the black squares represent the eigenvectors of $\boldsymbol{a}^{(2)}$

and the figure the associated eigenvalues. In the pole figures, the center of the circle corresponds to a co-latitude of zero. The co-latitude is $90^\circ$ and the azimuth is between $[0\ 2\pi]$ along the circle. In Fig. 16, we can visually observe that the majority of ice crystals cluster around the vertical axis with a small offset tilt angle.

From Eq. (9), the wave number difference between the ordinary and extraordinary waves in birefringence

ice is

$$k_2 - k_1 = \emptyset/(2z) \tag{16}$$

From Eq. (7) in [Hargreaves, 1977], the anisotropic dielectric permittivity $\epsilon_{z_p} - \epsilon_{x_p}$ is related to the wave number difference by

$$(\epsilon_{z_p} - \epsilon_{x_p})sin^2\beta = \frac{c^2}{\omega^2}\left(k_2{}^2 - k_1{}^2\right) \approx (k_2 - k_1)\,(2n/k_0) \tag{17}$$

5    where $n = 1.77$ is the ice refractive index and $k_0 = \pi$ /m is the wave number in air at 150 MHz. By substituting Eq. (16) into Eq. (17) we get

$$\beta = sin^{-1}\left[\sqrt{\frac{n\emptyset}{zk_0\left(\epsilon_{z_p} - \epsilon_{x_p}\right)}}\right] \tag{18}$$

Using the inferred birefringence phase shift from the radar multi-polarization measurements shown in Fig. 13a, for $\epsilon_{z_p} - \epsilon_{x_p} = 0.034$, the average angle of the optic axis with respect to the vertical axis at

10   different depths is estimated according to Eq. (18) and the result is presented in Fig. 17 at locations of Circle 1 to Circle 4. The mean value of all the points in Fig. 17 is $11.6°$ with a standard deviation of $0.8°$, and the minimum and maximum are about $10°$ and $14°$ respectively.

By applying the concept of the effective medium [Maurel et al., 2016], we calculate the effective co-latitude at 24 depths between 330 and 1025.8 m using the azimuth and co-latitude measurements of the

15   c-axis of ice crystals at these depths.  The orientation of the c-axis for a single crystal is represented by

$$\boldsymbol{c} = (c_1, c_2, c_3) = (sin\theta cos\varphi, sin\theta sin\varphi, cos\theta) \tag{19}$$

where $\varphi \in [0\ 2\pi]$ and $\theta \in \left[0\ \frac{\pi}{2}\right]$ are the azimuth and co-latitude respectively. The third component of the effective c-axis can be calculated by

$$c_3^{eff} = \int_0^{2\pi}\int_0^{\pi/2} p_n p(\varphi,\theta)cos\theta sin\theta d\theta d\varphi \tag{20}$$



where $p_n$ is a normalization factor and $p(\varphi, \theta)$ is the c-axis probability distribution function to meet the following normalization condition:

$$\int_0^{2\pi} \int_0^{\pi/2} p_n p(\varphi, \theta) \sin\theta d\theta d\varphi = 1 \tag{21}$$

Combining Eq. (20) and Eq. (21) we get:

$$c_3^{eff} = \frac{\int_0^{2\pi} \int_0^{\pi/2} p(\varphi, \theta) \cos\theta \sin\theta d\theta d\varphi}{\int_0^{2\pi} \int_0^{\pi/2} p(\varphi, \theta) \sin\theta d\theta d\varphi} \tag{22}$$

We also get:

$$\theta^{eff} = \cos^{-1}(c_3^{eff}) \tag{23}$$

Figure 18 displays an example of the measured c-axis orientation probability distribution as a function of $\varphi$ and $\theta$ at a depth of 330 m. The calculated effective co-latitudes are listed in Table 4, derived by

numerically integrating Eq. 22 with these measurements at the 24 depths (during the numerical integration, a step of $1°$ was used for both $d\varphi$ and $d\theta$). The average of these effective co-latitudes is $80.4°$ with a standard deviation of $1.2°$. This corresponds to an effective c-axis tilt angle of $9.6°$ from the vertical axis in very good agreement with the averaged angle $\beta = 11.6°$ between the optic axis and the vertical axis derived from the radar multi-polarization measurements. The co-latitudes at the 24 depths

are converted to the tilt angle to the vertical and overlapped in Fig. 17 (magenta circles) for a straightforward visual comparison with the previously estimated ice optic axis tile angle $\beta$. The minimum and maximum effective c-axis tilt angles from the vertical axis are ~$5.5°$ and ~$11.7°$ at 561 and 849.8 m respectively.

The effective co-latitude may not be physically and completely equivalent to the angle between the ice

optic axis and the vertical axis, but the agreement shown between them may provide some insights on how ice COF will affect the radio wave propagations in the medium. The orientation of the thin sections relative to the main direction of the ice core was controlled with an accuracy of about $1°$; the orientation

of the ice core by itself (relative to the vertical) is never well known and could vary by as large as $3^{\circ}$. These experimental errors and the limited number of sampled crystals may affect the agreement between the averaged $\beta$ and $\theta^{eff}$.

## 5 Summary, conclusions and discussion

In our multi-channel and polarimetric radar measurements performed around the NEEM ice core site in 2008, we identified a power reduction of about 12.7 dB transmitted towards the nadir in V-polarization because of the effects of the structure truss and mutual coupling on the antenna radiation pattern. Despite this issue, redundant HH and HV polarization measurements were successfully completed, providing a valuable dataset that reveals ice birefringence and COF characteristics. We presented a systematic method to process the data, effective for reducing speckle artefacts, enhancing the signal-to-noise ratio of ice layers, reducing power fading, and studying the bulk ice sheet birefringence and COF as a function of depth. The major findings from our analysis include:

(1) The comparison of the power variation patterns for HH and HV measurements with simulated theory-based patterns suggests that the ice COF at NEEM site is an elongated single maximum for depths between 450 and 1500 m, which agrees with the COF measurements from the ice core.

(2) The birefringence phase shifts between the ordinary and extraordinary waves increase nonlinearly as a function of depth.

(3) The ice optic axis lies in vertical planes that are approximately perpendicular to or align with the ice divide, depending on depth. The spatial variations of specific optic orientations are about $\pm 10^{\circ}$ across the 100-km$^2$ area around the NEEM site (see Fig. 19b). In general, the ice optic axis plane rotates slightly counterclockwise as depth increases.

(4) For the depth range between 324 and 1000 m, the ice optic axis has an average tilt angle of ~$11.6^{\circ}$ from the vertical axis, with a standard deviation of $0.8^{\circ}$ (see Fig. 19c). The average effective c-axis tilt angle from the vertical axis for 24 depths between 330 and 1025.8 is $\sim 9.6^{\circ}$ with a standard deviation





of 1.2˚. The agreement between the ice optic axis angle and effective c-axis tilt angle provides insight into how ice COF will affect the radio wave propagations in the medium.

(5) The anisotropic patterns and ice optical axis orientations revealed by the radar polarimetric measurements provide insight into the ice flow history of the NEEM site.

The orientation of the ice COF is dictated by forces in the ice column, and the preferred orientation of the girdle COF tends to be perpendicular to the direction of tension [van der Veen and Whillans, 1994]. The results of this investigation show two different girdle orientations within the area of study. In the case of ANISO-1 (from ~450 to ~1065 m),  the girdle COF is oriented perpendicular to the ice divide, indicating diverging ice flows along the ice divide (see Fig 19a, the main ice flow velocity vector is parallel to the
divide). Around the NEEM site, the main slope is in the divide direction, the slope is very low on both sides of the divide, and thus the flow scenario is divergent from the main flow along the divide (personal communication with Fabien Gillet-Chaulet). In the case of ANISO-2 (from ~1375 to 1575 m), the girdle COF is oriented parallel to the ice divide, suggesting the ice once flowed away from the ice divide on both sides in a direction orthogonal to the ice divide. The presence of both COF orientations shows the
complex flow history of the area of study and demonstrates the need for additional investigations in both present and previous ice flows of the region.

*Acknowledgements.*   This project was supported by NSF grant ANT-0424589 and NASA grant NNX16AH54G. The authors would like to thank all the students and staff at CReSIS who took part in the
development of the radar system and the data collection in the field. They also acknowledge the support from the NEEM team during the data collection and particularly Sepp Kipfstuhl for conducting most COF measurements. IW acknowledges funding by HGF VH-NG-802.  R. M. James is acknowledged for editing the manuscript.



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





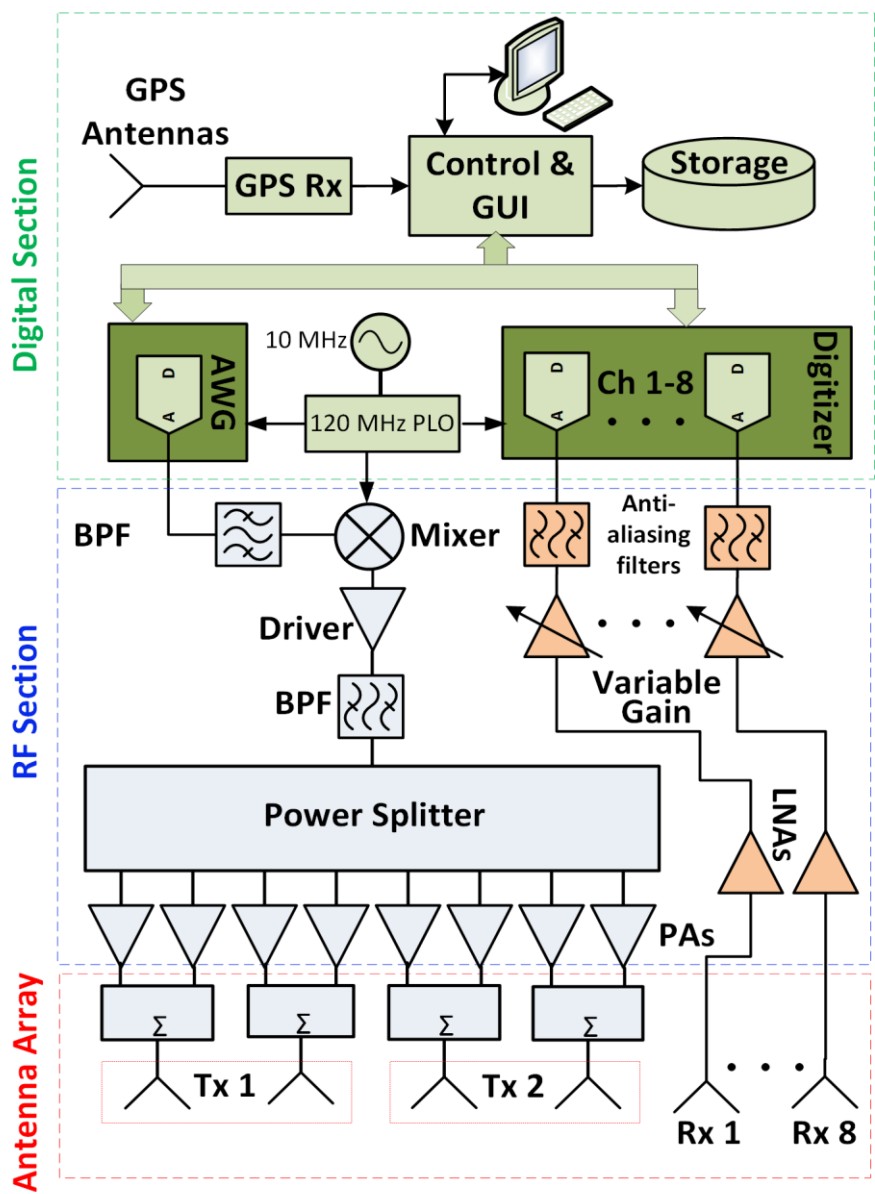

**Figure 1: The simplified block diagram of the multi-channel radar system.**





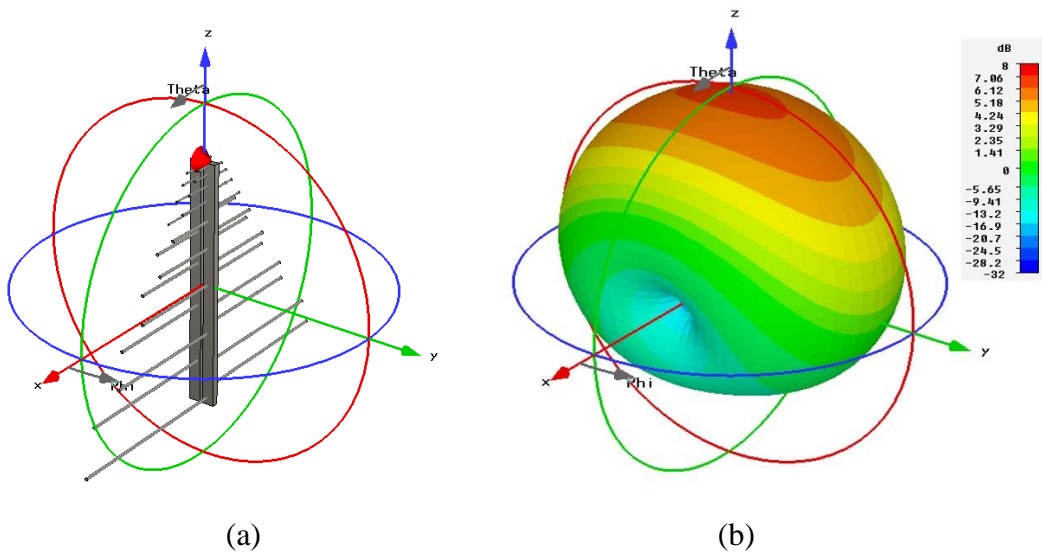

(a)            (b)

**Figure 2: The geometry and radiation patterns of the 17-element log-periodic antenna. The length of the longest dipole is 1.15 m and the spacing between the shortest dipole and the longest dipole is 0.75 m. The radiation pattern was generated by using the 3D electromagnetic simulation software CST (Computer Simulation Technology) Microwave Studio.**

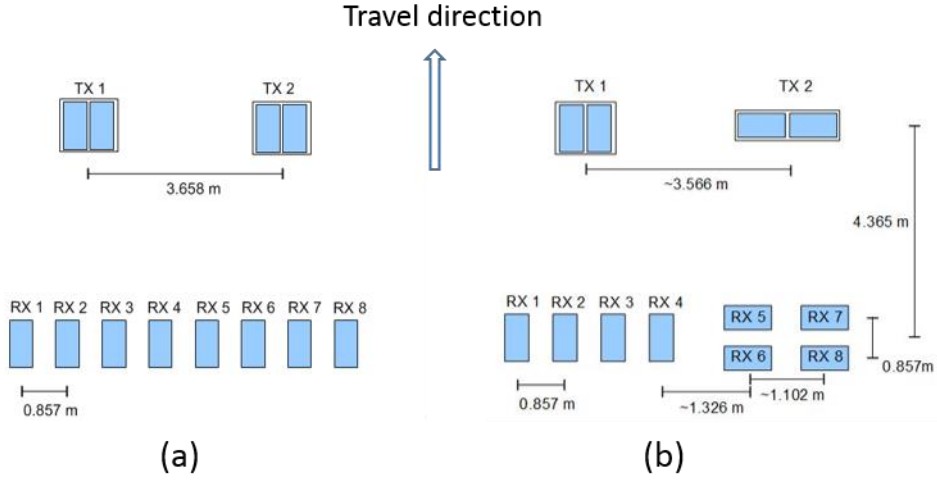

**Figure 3: Antenna configurations: (a) non-polarimetric HH polarization setup; (b) quad-polarimetric setup.**




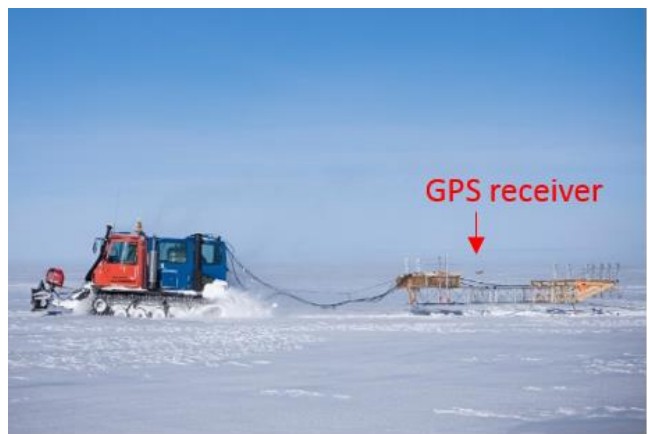

**Figure 4: Multi-polarization data collection with quad-polarimetric antenna setup. Left: antennas on the sled towed by a snow tractor. Right: H and V receive antennas.**

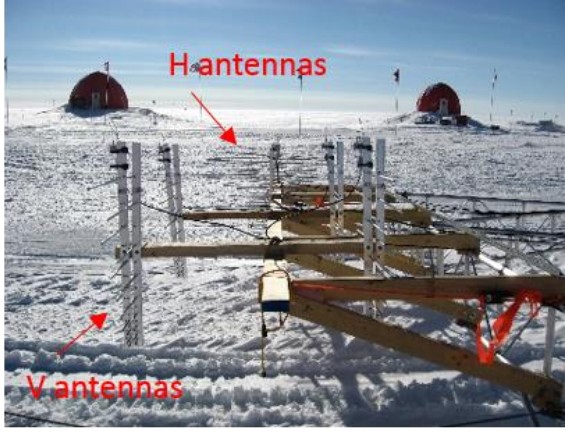
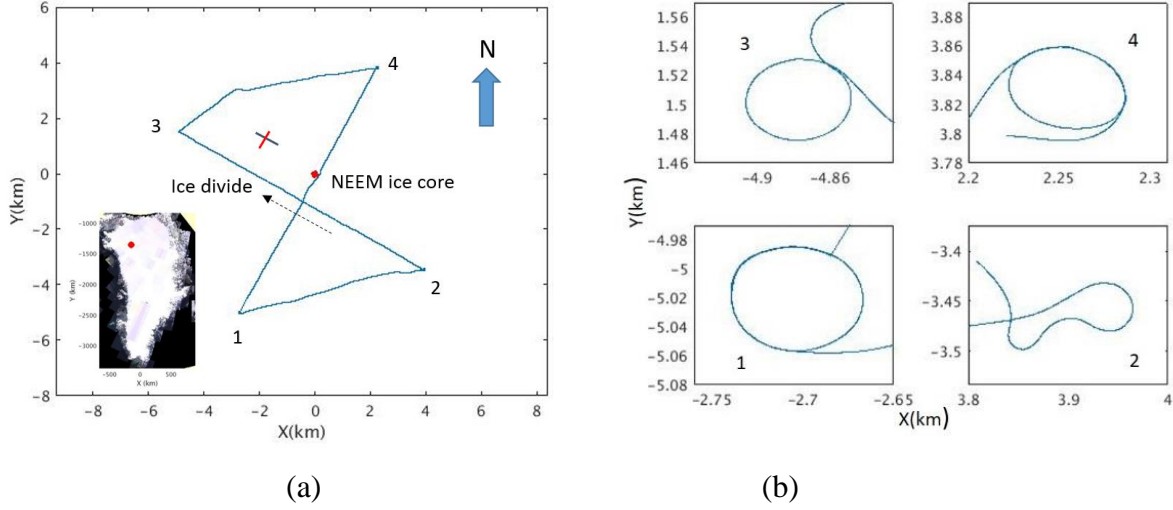

(a)                                          (b)

**Figure 5: Paths of multi-polarization measurements: (a) Paths along ice divide and perpendicular to ice divide. (b) Circular paths at turns. The paths are plotted in the local ENU (East-North-Up)**
10   **coordinate system at the NEEM ice core camp site (77.45°N 51.06°W), marked by a red dot at (0,0). The red dot in the inset map shows the location of the NEEM site in Greenland. The cross formed by a red and a blue segment shows the location of a crossover where the data were collected with the non-polarimetric HH polarization antenna configuration (see discussions on Fig. 14).**



**Figure 6: Averaged power-depth profile along straight line perpendicular to the ice divide (from pt. 1 to pt. 4 in Fig. 5).**



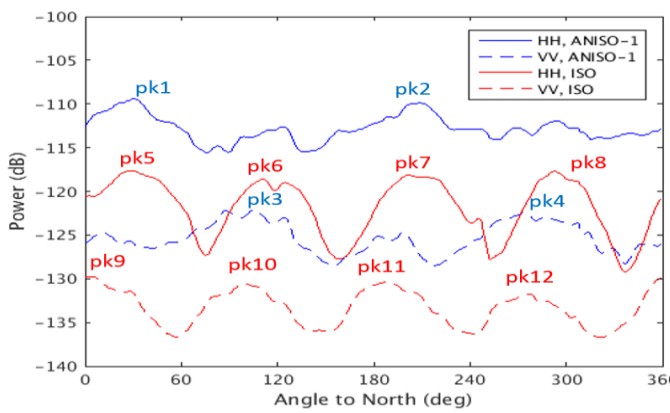

**Figure 7: Power variation pattern of HH (TX1-RX4) and VV (TX2-RX5) along Circle 3. Blue lines represent the anisotropic zone ANISO-1 (see Fig. 14 a and b) at 840-m depth, red lines represent the isotropic zone at 1178-m depth, solid lines represent HH measurements and dashed lines represent VV measurements.**

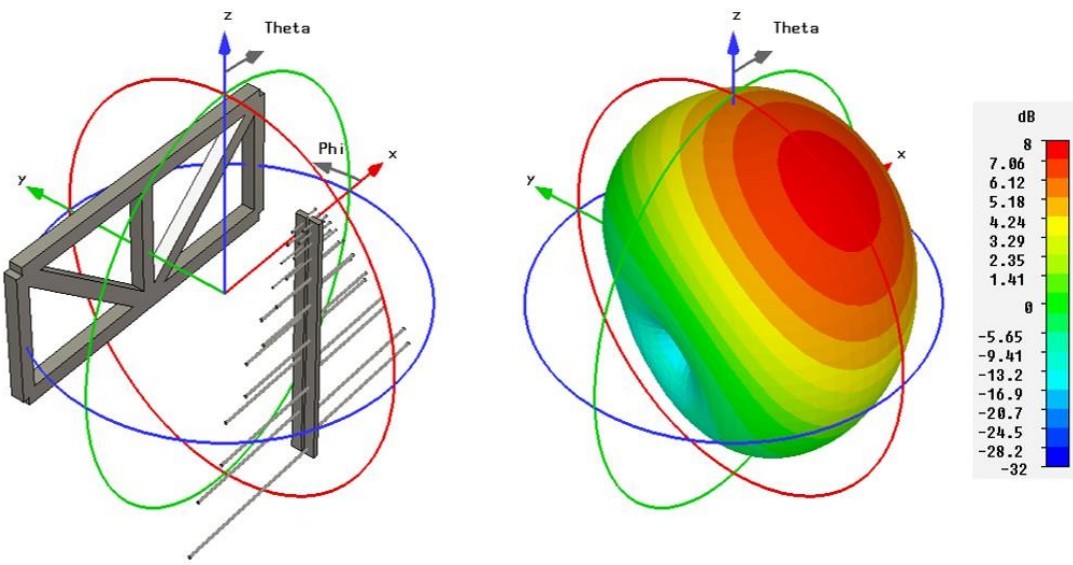

10   **Figure 8: Truss effect on antenna radiation pattern.**





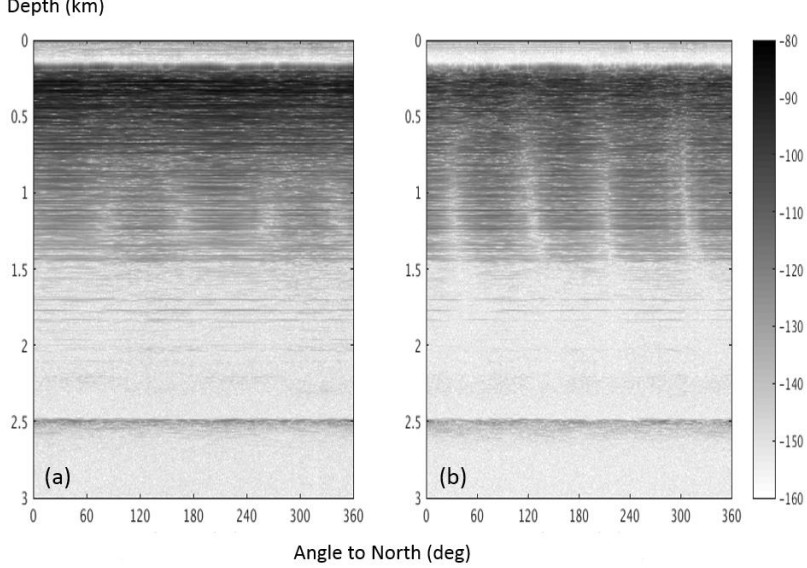

**Figure 9: Multi-polarization measurements along Circle 3: (a) HH, transmit TX1, receive channels RX1, RX2, RX3 and RX4 combined; (b) HV, transmit TX1, receive channels RX5, RX6, RX7 and RX8 combined. The horizontal axis is the angle in degrees with respect to the local north, the vertical axis is ice depth in kilometers and the color bars represent the relative power level in decibels.**

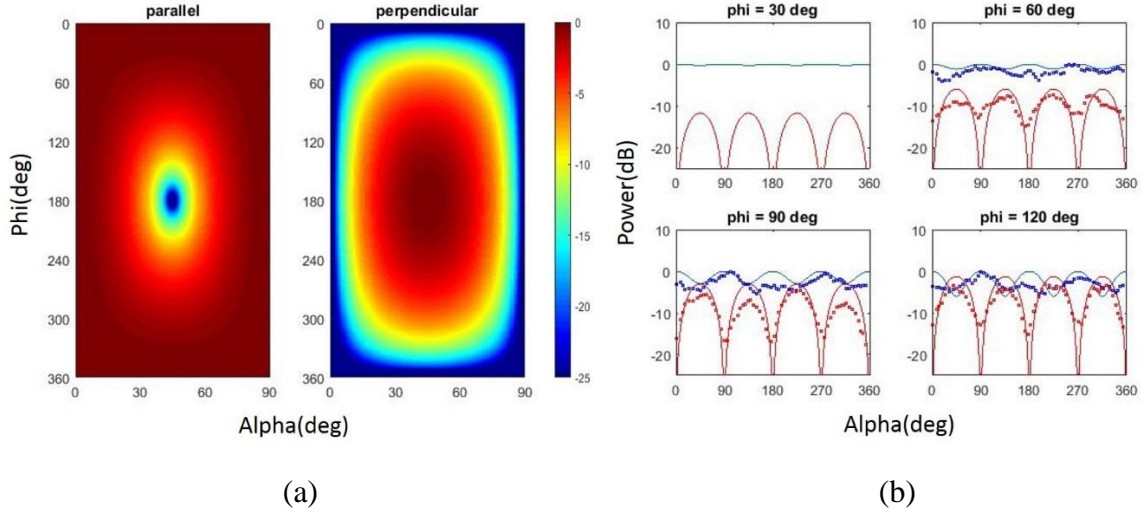

(a)                                                                  (b)

**Figure 10: Effects of the angle α between TX1 and x-axis and birefringence phase shift ∅ on reflected power. The color bar in (a) gives the power scale in the range from 0 dB to -25 dB. The blue and red lines correspond to co-polarization and cross-polarization respectively. The solid lines are simulated and dotted lines are measured.**



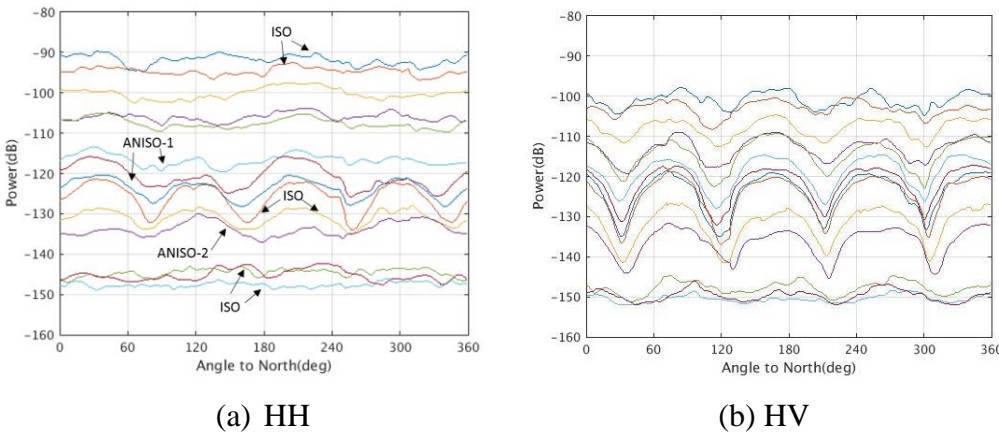

(a) HH                                    (b) HV

**Figure 11: Power profiles with respect to the angle to North at different depths (see the caption of Fig. 14 about the annotations in (a)).**

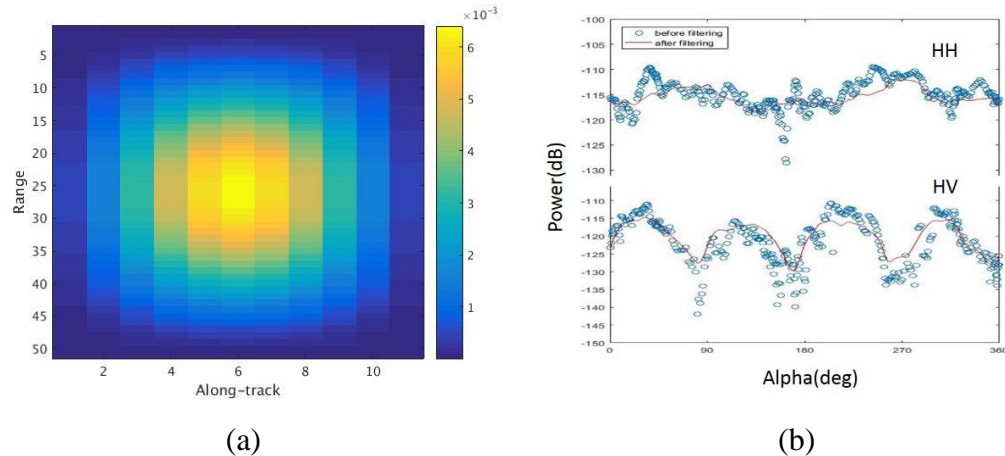

(a)                                        (b)

**Figure 12: A 2-D moving average filter applied to data: (a) the filter kernel image; (b) comparison between data before and after filtering.**





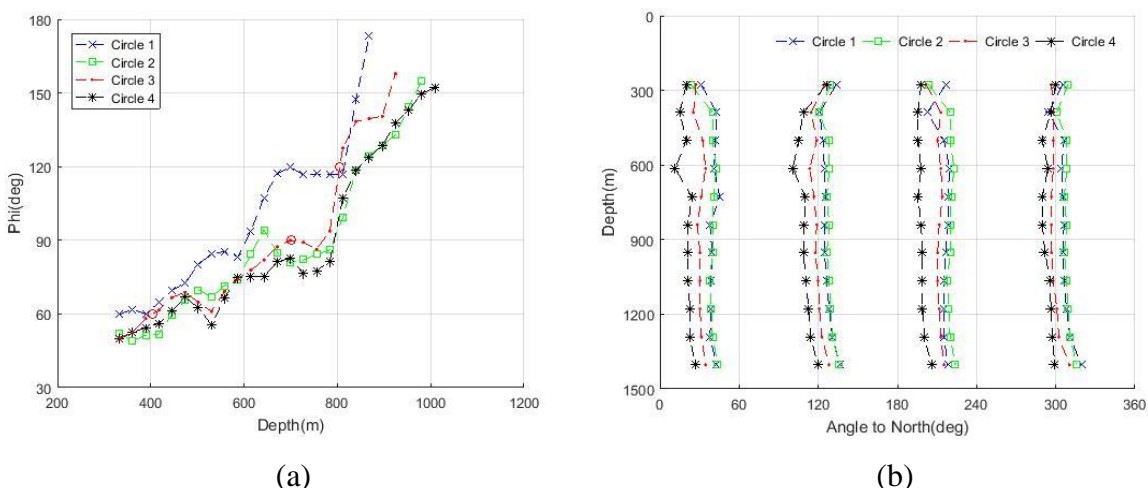

(a)                                                          (b)

**Figure 13: (a) Estimated birefringence phase shift as a function of depth; (b) HV power minimum locations. The ice divide is parallel to $119.3°$ (and $299.3°$) in this figure.**

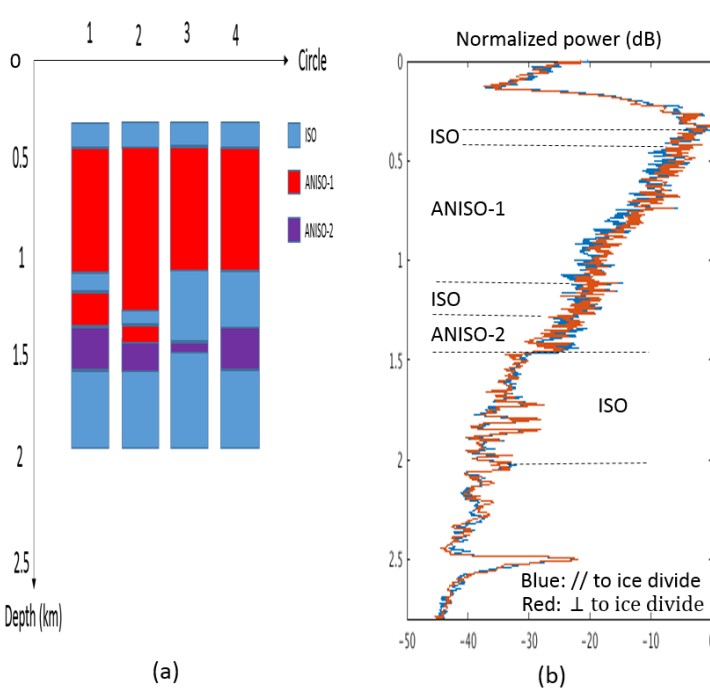





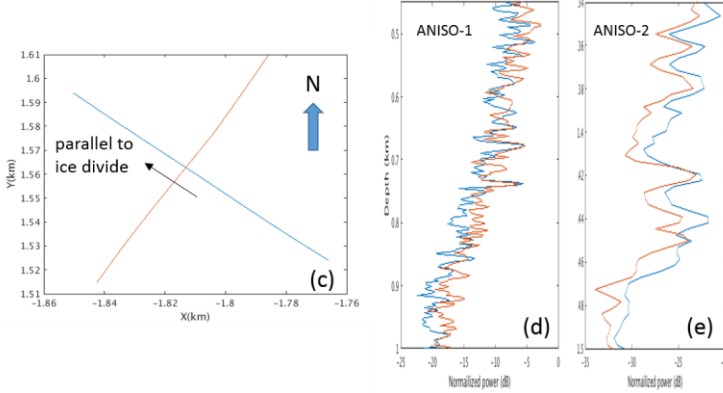

**Figure 14: (a) Power variation patterns and their depth profiles for the HH polarimetric measurements at the four circles. ISO indicate no obvious anisotropy observed. ANISO-1/ANISO-2 are anisotropic patterns and indicate the ice optic axis is perpendicular/parallel to the ice divide.**

5  **(b) Stacked normalized power-depth profiles at a crossover from non-polarimetric measurements with HH antenna configuration. (c) Paths of the crossover (see the cross mark in Fig. 5a for its location relative to the four circles). (d) Anisotropic pattern ANISO-1 at the crossover. (e) Anisotropic pattern ANISO-2 at the crossover.**



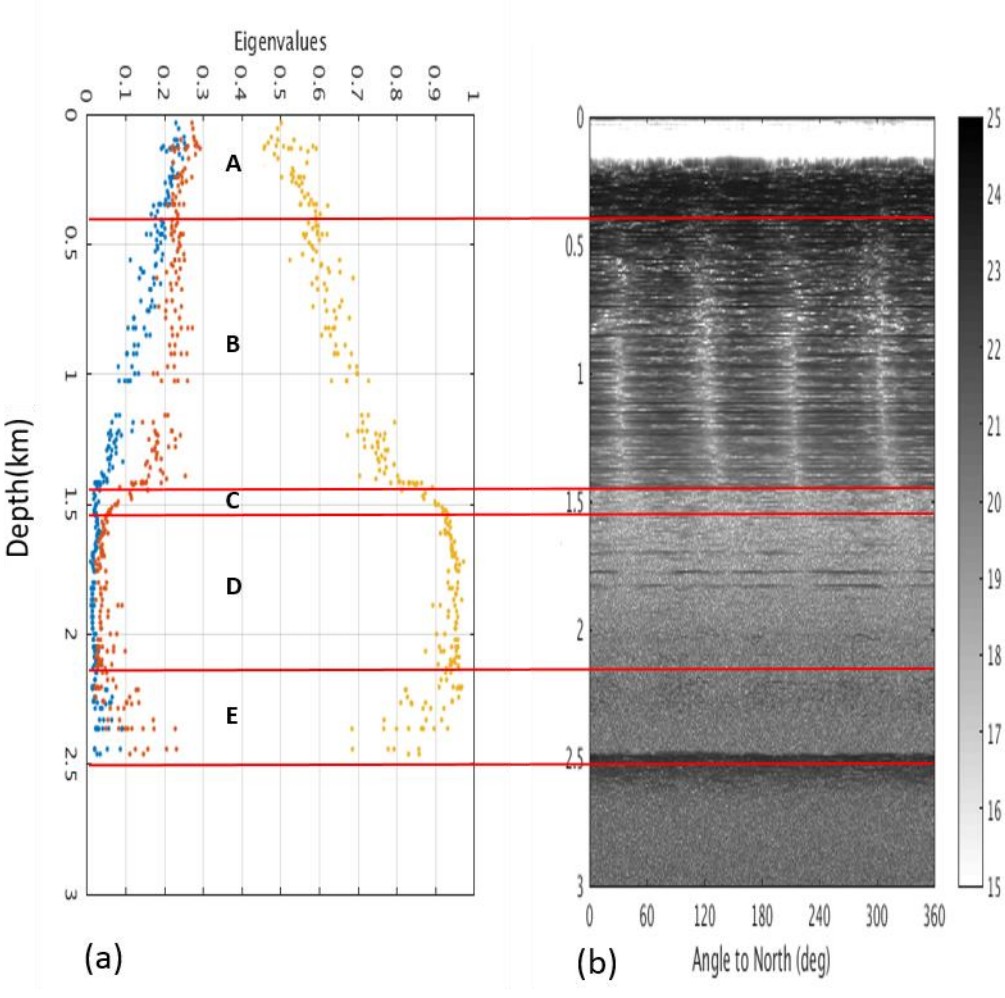

**Figure 15: (a) Fabric profile along the NEEM ice core represented by the eigenvalues of the second-order orientation tensor (yellow: $a_1^{(2)}$, red: $a_2^{(2)}$, and blue: $a_3^{(2)}$) [Eichler et al., 2013; Montagnat, et al., 2014]; (b) Radar echogram of HV measurements at Circle 3, adopted from Fig. 9b for visual comparison with the ice core COF. A—single max; B—elongated single max; C—transition from B to D; D—strengthened single max; E—folded ice.**



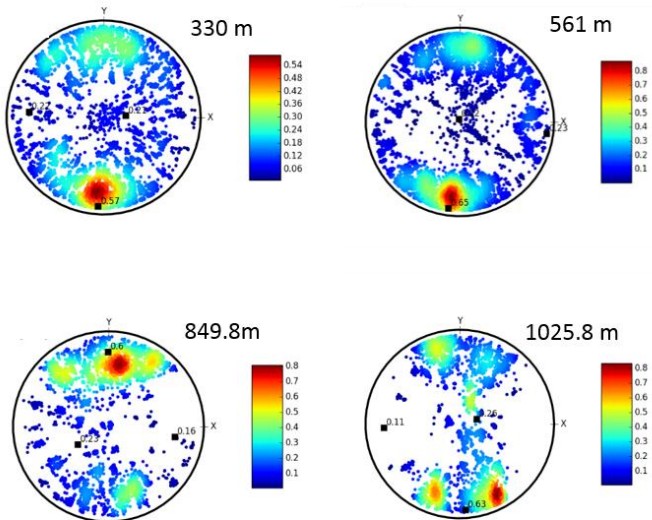

**Figure 16: Fabric represented in stereographic projection/pole figure projected into the thin section plane at depths of 330, 561, 849.8 and 1025.8 m. The *OY* direction in the pole figure is approximately the vertical ice core axis. The center of the diagram corresponds to the approximate horizontal.**
5 **Only 10,000 data points randomly selected are plotted for each thin section. The color bar corresponds to the density of plotted data points.**

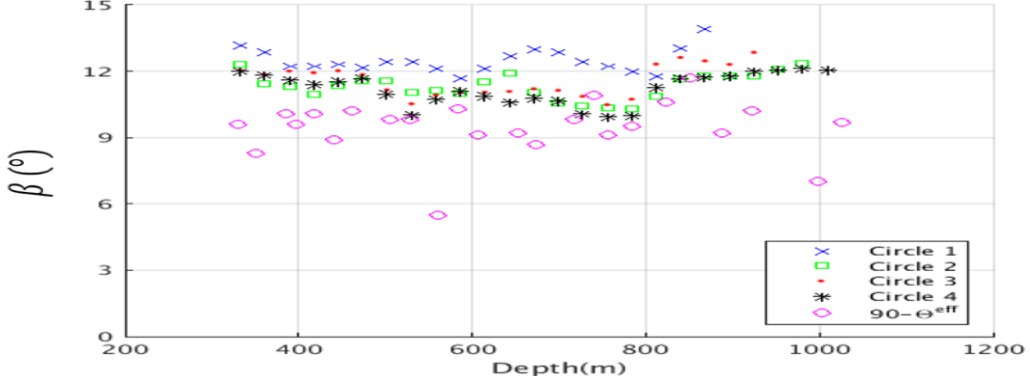

**Figure 17: Comparison between the average angles of optic axis estimated from multi-polarization**
10 **measurements and the effective c-axis angles at different depths.**



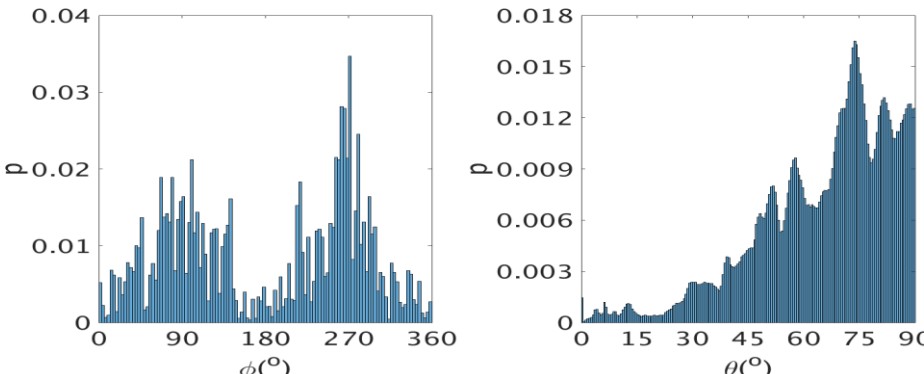

**Figure 18: Measured c-axis orientation distributions as a function of φ and θ at depth 330 m. Any measurements with a quality factor of less than 70 were disregarded and not included in the distribution plots (see [Peternell et al., 2010] for details about the quality factor). In total, there are qualified 2032533 crystal c-axis orientation measurements at this depth.**

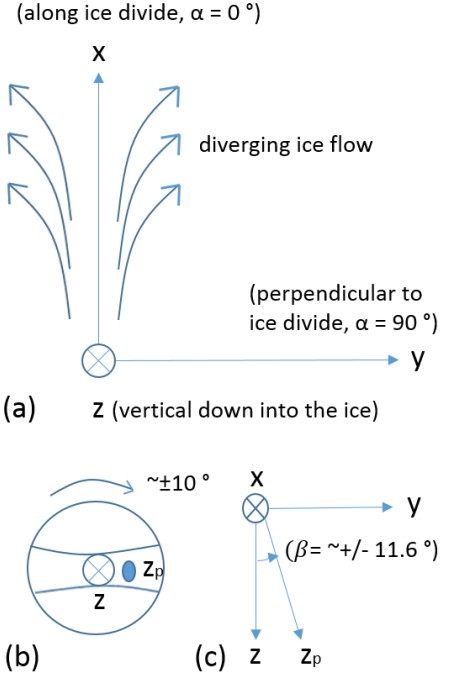

**Figure 19: Illustration of the major findings from the multi-polarization radar measurements. (a) Diverging ice flow scenario at ice divide; (b) Sketch of girdle COF formed by diverging ice flow in which the point $z_p$ is the projection of the ice optic axis, and rotation arrow indicates the spatial variations of the ice fabric orientation around the NEEM site; (c) The average tilt angle of ice optical axis $z_p$ at the NEEM site.**



**Table 1: System parameters**

| Parameter | Value | Units |
|---|---|---|
| Frequency band | 135 to 165 | MHz |
| Pulse Duration | 3 or 10 | µs |
| Pulse Repetition Frequency | 10 | kHz |
| Transmit Power | 400 | W |
| Receiver Noise Figure | 3.9 | dB |
| Loop Sensitivity | 212 | dB |
| Minimum Detectable Signal | -162 | dBm |
| Range Resolution | 2.8 | m |
| Coherent Averages | 64 | |
| Antenna Element | log-periodic | |
| Antenna Gain | 6.7 | dB |
| A/D Converter | 12 | bit |
| Sampling Rate | 120 | MSPS |

**Table 2: Time delay $\nabla t_d$ , amplitude $\Delta A$ and phase $\Delta\emptyset$ mismatches between channels**

| | RX1, RX2, RX3, RX4 | RX5, RX6, RX7, RX8 |
|---|---|---|
| $\nabla t_d$(ns) | -2.52, 0, -0.55, 1.06 | -0.79, 0, -1.24, -7.09 |
| $\Delta A$(dB) | 2.8, 0, 1.8, 2.9 | 2.1, 0, 1.8, 0.4 |
| $\Delta\emptyset$(°) | -162.3, 0, -91.5, -19.5 | -23.9, 0, -105.6, 41.9 |



**Table 3: Received power (in dB) of different polarizations from specular layer at depth of 1142m**

|  | RX4 (H) | RX5 (V) |  |
|---|---|---|---|
| TX1 (H) | $P_{HH}$ = -107 | $P_{HV}$ = -115.5 | $\Delta Pc = P_{HH} - P_{HV} = 8.5$ |
| TX2 (V) | $P_{VH}$ = -123.4 | $P_{VV}$ = -121.2 | $\Delta Pd = P_{VH} - P_{VV} = -2.2$ |
|  | $\Delta Pa = P_{HH} - P_{VH} = 16.4$ | $\Delta Pb = P_{HV} - P_{VV} = 5.7$ |  |

**Table 4: Effective co-latitudes $\theta^{eff}$ (°) calculated from COF measurements at different depths**

| Depth(m) | 330 | 352 | 385 | 396 | 418 | 440 | 462 | 506 | 528 | 561 | 583 | 607.3 |
|---|---|---|---|---|---|---|---|---|---|---|---|---|
| $\theta^{eff}$ (°) | 80.4 | 81.7 | 79.9 | 80.4 | 79.9 | 81.1 | 79.8 | 80.2 | 80.2 | 84.5 | 79.7 | 80.9 |
| Depth(m) | 652.4 | 674.4 | 717.8 | 739.8 | 755.8 | 783.8 | 822.3 | 849.8 | 888.8 | 921.8 | 998.2 | 1025.8 |
| $\theta^{eff}$ (°) | 80.8 | 81.3 | 80.2 | 79.1 | 80.9 | 80.5 | 79.4 | 78.3 | 80.8 | 79.8 | 83.0 | 80.3 |

