# Peer review of "Multi-Channel and Multi-Polarization Radar Measurements around the NEEM Site"

_The Cryosphere, 2018_

## Referee Comment (RC1) · H. Conway (Referee) · 2 Jun 2018

Others have shown that depth-profiles of radar-detected fabric show good agreement with direct measurements of fabric from ice cores (e.g. Drews et al. 2012, Eisen et al., 2007 at EDML; Fujita et al, 2006 at Dome Fuji and Mizuho). Here, the authors report multi-channel and multi-polarization radar measurements in the vicinity of the NEEM ice-core site. The radar measurements are used to infer depth-profiles of birefringence and fabric and compared with published measurements along the core (Montagnat et al. 2014 and Weikusat et al. 2017).

The new measurements, together with those from previous work, gives confidence that profiles of ice fabric in ice sheets can be determined using remote sensing methods.

[Figure]

Given the importance of ice-fabric on ice dynamics, and the logistical difficulties of extracting and processing ice cores, this is an important contribution.

A few comments/questions: Sections 2, 3.1 & 3.2 provide in-depth descriptions and details of a very impressive radar system, data collection and processing methods. You also note a power mismatch between the two transmitters when used in HH and VV orientations, and provide a simulation to estimate the influence of the truss on the antenna radiation pattern. Conclusion is that the power mismatch was likely caused by interference between the radiation patterns. The focus for the remaining part of the paper is on polarization measurements from just one of the transmitters, which includes HH and HV transmit-receive polarizations. In order to make the manuscript more accessible to readers who are not so interested in details of the radar system, I suggest you consider moving these detailed descriptions (together with Figs. 1, 2, 3, 4, 6, 7 & 8) into supplementary information.

In order to keep the manuscript accessible to readers, you might also consider moving the discussion of filtering data (together with Figs 11&12) to the supplementary section.

Detailed questions

Page 1 - line 29. I think this point should be the main emphasis of the abstract. It should be expanded to state details of the "very good agreement". The importance of the work is that it enables confidence in radar-detected polarimetry and it can be applied in places where there are not measurements from ice cores.

Page 10 - line 5ff. Does this mean that you profiled along circle 3 twice, or something else? - line 9. In fig.9 it looks like the upper ~125m is blank (rather than 250m expected from a 3us delay). - line 16. Do you mean signal "extinction" (rather than "distinction")?

Page 13 - line 21ff. Reference to Fig.10b -it would be helpful to also explain Fig. 10b in the figure caption (Page 34). Are dotted "measured" lines raw data or have they been filtered? Are they from Circle 3 or from some other circle?

Page 15 - line 5ff. I have trouble interpreting Fig. 11. What do the different colors of power profiles represent? It is not clear to me what criteria has been used to assign the annotations in Fig. 11a. I see now that this is described on Page 17 line18ff. It would help if you pointed the reader to this explanation. Also, it would be helpful to include a note about the annotations in the caption rather than having to refer to Fig. 14.

Page 18 - Concerning Figs 14b & 14d; how is normalized power determined in the upper 250m? Also, the depth scale in Fig. 14e needs to be clear; I am not sure – does it range from 1.34 to 1.5km?

Page 19 - lines 11&14. What is meant by "at the internal ice layer interface"? It might also be instructive to plot a full-depth-profile of the power difference, which might help the reader to delineate regions of ISO/ANISO-1/ANISO-2. - line 21. Fig 15 – not 14. It might be a more instructive comparison to compare the measured profile of eigen values with a full-depth-profile of the power difference, calculated either from Fig.14b or from the cross-over measurements (Fig. 14d&e)

Page 20 - line 9. Presumably you observe "weak radar reflectors" (rather than "weak ice layers") - line 18. Fig. 16 seems redundant; full-depth information is given in Fig 15a

Page 24 - line 3. Is this correct? I suspect that the borehole measurements of fabric have already provided insight into the ice flow history. More important here is that the radar polarimetry measurements are closely similar to measurements from the borehole and can be used with confidence to extrapolate ice-flow histories spatially. - line 5. Do you mean stresses rather than forces?

---

## Author Comment (AC1) · 19 Jun 2018

Dear Prof. Conway,

Thank you very much for reviewing the manuscript and for your valuable comments to improve our manuscript. We would like to first address your comments/questions (in red), and then integrate them into the final manuscript:

Sections 2, 3.1 & 3.2 provide in-depth descriptions and details of a very impressive radar system, data collection and processing methods. You also note a power mismatch between the two transmitters when used in HH and VV orientations, and provide a simulation to estimate the influence of the truss on the antenna radiation pattern. Conclusion is that the power mismatch was likely caused by interference between the radiation patterns. The focus for the remaining part of the paper is on polarization measurements from just one of the transmitters, which includes HH and HV transmit-receive polarizations. In order to make the manuscript more accessible to readers who are not so interested in details of the radar system, I suggest you consider moving these detailed descriptions (together with Figs. 1, 2, 3, 4, 6, 7 & 8) into supplementary information.

We will move this into supplementary documents, as suggested.

In order to keep the manuscript accessible to readers, you might also consider moving the discussion of filtering data (together with Figs 11&12) to the supplementary section.

We will make this change.

Page 1 - line 29. I think this point should be the main emphasis of the abstract. It should be expanded to state details of the "very good agreement". The importance of the work is that it enables confidence in radar-detected polarimetry and it can be applied in places where there are not measurements from ice cores.

We expanded the abstract as follows (added text is marked in green):

**Abstract.** Ice properties inferred from multi-polarization measurements, such as birefringence and crystal orientation fabric (COF), can provide insight into ice strain, viscosity and ice flow. In 2008, the Center for Remote Sensing of Ice Sheets (CReSIS) used a ground-based VHF radar to take multi-channel and multi-polarization measurements around the NEEM (North Greenland Eemian Ice Drilling) site. The system operated with 30 MHz bandwidth at a center frequency of 150MHz. This paper describes the radar system, antenna configurations, data collection, and processing and analysis of this data set. In the area of $100 \text{ km}^2$ around the ice core site, ice birefringence dominates the power variation patterns of co-polarization and cross-polarization measurements. The phase shift between the ordinary and extraordinary waves increases nonlinearly with depth. The ice optic axis lies in planes that are close to the vertical plane and perpendicular or parallel to the ice divide depending on depth. The ice optic axis has an average tilt angle of about $11.6^{\circ}$ vertically, and its plane may rotate either clockwise or counterclockwise by about $10^{\circ}$ across the $100\text{-km}^2$ area, and at a specific location the plane may rotate slightly counterclockwise as depth increases. Comparisons between the radar observations, simulations, and ice core fabric data are in very good agreement. We calculated the effective co-latitude at different depths by using azimuth and colatitude measurements of the c-axis of ice crystals. We obtained an average effective c-axis tilt angle of 9.6° from the vertical axis, which is very comparable to the average optic axis tilt angle estimated from the radar polarization measurements. The comparisons give us confidence in applying this polarimetric radio echo sounding technique to infer profiles of ice fabric in locations where there are no ice core measurements.

Page 10 - line 5ff. Does this mean that you profiled along circle 3 twice, or something else?

No, we did not profile twice along Circle 3. The two coherent integrations means using the average of every two consecutive data records of complex values to replace the original two data records. It decimated the number of data records by a factor of two.

- line 9. In fig.9 it looks like the upper _125m is blank (rather than 250m expected from a 3us delay).

Yes, there is a factor of 2 after considering the two-way propagation. Thanks for the correction. We revised it as follows:

In Fig. 9, the top ~127 m is blank. This corresponds to the eclipsing time, which is equal to the duration of the 3-µs chirps that the radar has to wait before recording data for unambiguous range detection. … As seen in Fig. 9, the receive power level obtained for HH measurements is higher than that of HV measurements for shallow depths between 127 and 750 m, and the power level of HV measurements becomes comparable to that of HH measurements for depths of 750 to 1450 m.

- line 16. Do you mean signal "extinction" (rather than "distinction")?

Yes, it should be "extinction". We have corrected this typographic error.

Page 13 - line 21ff. Reference to Fig.10b -it would be helpful to also explain Fig. 10b in the figure caption (Page 34). Are dotted "measured" lines raw data or have they been filtered? Are they from Circle 3 or from some other circle?

Thanks for the suggestion. We revised the caption of Fig.10b as follows:

**Figure 10: Effects of the angle α between TX1 and x-axis and birefringence phase shift ∅ on reflected power. The color bar in (a) gives the power scale in the range from 0 dB to -25 dB. The blue and red lines correspond to co-polarization and cross-polarization respectively. The solid lines are simulated and dotted lines are filtered measurements along Circle 3.**

Page 15 - line 5ff. I have trouble interpreting Fig. 11. What do the different colors of power profiles represent? It is not clear to me what criteria has been used to assign the annotations in Fig. 11a. I see now that this is described on Page 17 line18ff. It would help if you pointed the reader to this explanation. Also, it would be helpful to include a note about the annotations in the caption rather than having to refer to Fig. 14.

The purpose of Fig. 11 is to show the bulk properties of the received power at different depths. In Fig. 11, there are 14 HH and VV power profiles, respectively, at 14 depths. The colors are automatically determined by MATLAB's plot command, which cycles through its default colors. To avoid ambiguity, we revised Fig. 11 by using four colors (blue, red, green and magenta) to distinguish the profiles that are close to each other and by labelling the different profiles with numbers (from 1 to 14 in increasing order) according to the depths from the top to the bottom. The updated plots are included below. To make it easier for readers to understand the annotations in Fig. 11a, we revised the caption as suggested.

[Figure]

**Figure 11: Power profiles with respect to the angle to North at different depths** **for HH and HV measurements. Four colors (blue, red, green and magenta) are cycled through for distinguishing close profiles and all the profiles are numbered from 1 to 14 in increasing order, according to their respective depths from the top to the bottom. The annotations in (a) illustrate the isotropic and anisotropic patterns discussed with Fig. 14 at the end of Section 3.3.**

Page 18 - Concerning Figs 14b & 14d; how is normalized power determined in the upper 250m? Also, the depth scale in Fig. 14e needs to be clear; I am not sure – does it range from 1.34 to 1.5km?

Thanks for pointing out the ambiguities. In Fig. 14b & Fig. 14d, the power profiles parallel-to-ice-divide (in blue) and perpendicular-to-ice-divide (in red) were normalized by dividing them by the maximum of the parallel-to-ice-divide power profile, which appears at depth of ~327 m. In Fig. 14e, the depth scale ranges from 1.34 to 1.5 km. We revised Fig. 14e and its caption as shown below:

[Figure]

Figure 14: (a) Power variation patterns and their depth profiles for the HH polarimetric measurements at the four circles. ISO indicates no obvious anisotropy observed. ANISO-1/ANISO-2 are anisotropic patterns and indicate the ice optic axis is perpendicular/parallel to the ice divide. (b) Stacked normalized power-depth profiles and their differences at a crossover from non-polarimetric measurements with HH antenna configuration. The normalization is done by dividing both profiles by the maximum of the blue one at 327 m. The plot of the differences is area-filled. (c) Paths of the crossover (see the cross mark in Fig. 5a for its location relative to the four circles). (d) Anisotropic pattern ANISO-1 at the crossover. (e) Anisotropic pattern ANISO-2 at the crossover.

Page 19 - lines 11&14. What is meant by "at the internal ice layer interface"?

Because the power difference maxima appear at local peaks, when we said "at the internal ice layer interface" we meant "at the internal ice layer". We revised the text as follows:

…with a maximum difference of 6.3 dB at the depth of ~462 m … with a maximum difference of 4.4 dB at the depth of ~1459 m.

It might also be instructive to plot a full-depth-profile of the power difference, which might help the reader to delineate regions of ISO/ANISO-1/ANISO-2.

We revised Fig. 14b to include the full-depth-profile of the power difference and the caption was updated accordingly, as follows:

[Figure]

(b)

Figure 14: (a) Power variation patterns and their depth profiles for the HH polarimetric measurements at the four circles.  ISO indicates no obvious anisotropy observed. ANISO-1/ANISO-2 are anisotropic patterns and indicate the ice optic axis is perpendicular/parallel to the ice divide. (b) Stacked normalized power-depth profiles and their differences at a crossover from non-polarimetric measurements with HH antenna configuration. The normalization is done by dividing both profiles by the maximum of the blue one at 327 m. The plot of the differences is area-filled. (c) Paths of the crossover (see the cross mark in Fig. 5a for its location relative to the four circles). (d) Anisotropic pattern ANISO-1 at the crossover. (e) Anisotropic pattern ANISO-2 at the crossover.

- line 21. Fig 15 – not 14.

We corrected this typo both at line 21 & 24. Thanks for pointing out it.

It might be a more instructive comparison to compare the measured profile of eigenvalues with a full-depth-profile of the power difference, calculated either from Fig.14b or from the cross-over measurements (Fig. 14d&e)

Please see above regarding revisions to Fig. 14b.

Page 20 - line 9. Presumably you observe "weak radar reflectors" (rather than "weak ice layers")

We revised the text and changed "weak ice layers" to "weak reflectors".

- line 18. Fig. 16 seems redundant; full-depth information is given in Fig 15a

We consider that that Fig. 16 may be supplementary, but not redundant, as it displays the raw ice core data from which the eigenvalues of Fig. 15a were derived. It is also a complement to Fig. 18 from which the effective co-latitude at different depths were calculated. We would be happy to change to supplementary material if recommended, but we would definitely like to keep it.

Page 24 - line 3. Is this correct? I suspect that the borehole measurements of fabric have already provided insight into the ice flow history. More important here is that the radar polarimetry measurements are closely similar to measurements from the borehole and can be used with confidence to extrapolate ice-flow histories spatially.

We argue that the orientation of the crystals in the core is only known relative to the vertical axis. The orientation of the core in the x-y plane after it was extracted is not known. Therefore, direct checks of ANISO-1 and ANISO-2, which provide insight to ice flow, are not possible with core data. We agree with the referee's statement that the radar polarimetry measurements can be used with confidence to extrapolate ice-flow histories spatially, therefore we add the following hereafter:

(6) The very good agreement between the radar observations, simulations, and ice core fabric data provides assurance as to the effectiveness of polarimetric radio echo sounding techniques to infer profiles of ice fabric in locations where no ice core data are available.

-line 5. Do you mean stresses rather than forces?

Yes. We revised "forces" as "stresses".

---

## Referee Comment (RC2) · Z. Courville (Referee) · 21 Jun 2018

General Comments: This is a general well-written manuscript that presents very interesting results comparing multi-channel/multi-polarization radar results to measured crystal orientation fabric from the NEEM ice core. The results show good comparison between the two methods. The methods are well-presented and clear.

Specific Comments: Page 5, Line 6: need more technical information about the GPS receiver Page 9, line 11: The truss, since it has such a large impact on the transmit power mismatch, should be introduced and explained in more detail (geometry? Construction? Materials?) in Section 2. Page 19, section 4: suggest adding a brief description of the COF methods, for instance, in Figure 16 there is mention of thin

sections. It would probably be good to mention that the COF are obtained from thin sections in the text. (or in the caption of Figure 16, instead of saying "projected into the thin section plane" name what that plane (guessing that it is vertical, but actually not sure). Page 23, line 7: Earlier (Page 9, line 13), there is mention that the power reduction is due to these two factors as well as other unknown factors. That there are unknown factors contributing to the power reduction should also be mentioned in the summary.

Technical Comments: Throughout: don't need a comma after the name and before "et al." in citations (i.e., in lines 10 and 11), but do need a comma after et al. (i.e., line 17) Page 1, Line 19: define VHF Page 3, Line 15: Why is km bold and italicized? Page 3, Line 24: should be, "plane also rotates.." (or "the planes also rotate..") if this is referring to the vertical planes. Page 4, Line 7: should be comma after "Marathe" in citation Page 5, line 4: "snow tractor" is not a term I've heard before. We typically call these "tracked vehicles." Page 5 and throughout: should be consistent if there is a space after period in Fig or not (i.e., there is a space on page 4, Line 25 after Fig. 3, but not on lines 4 and 5 on page 5. Page 6, line 22: why is the m in bold? Page 8, several instances: why is dB in parentheses here, and not above? Should be consistent at least. Page 9, line 9: Fig. 7 should be capitalized Page 11, line 4: seems like you should say something along the lines of, " The plane wave is of linear, circular, or elliptical polarization when this curve is, respectively, a straight line, a circle, or an ellipse, [Ulaby, 1981]." To make this sentence clearer. Page 17, line 15: do the authors mean, optic axis instead of optical axis here? Page 19, line 21: think that this is actually supposed to be referring to Figure 15, not Figure 14. Page 21, line 4: The variable beta should be defined. Page 22, line 21: This is the first mention of thin sections in the text. I think there should be a brief discussion of the methods on Page 16, or refer to these as "samples," but be consistent. Page 22, line 16: "tile" should be "tilt"

Figure 2: Maybe add a label with the dipole lengths in the figure? Figure 5: Better label the ice divide? Right now, it looks like the "Ice divide" label is associated with the

arrow, which I think is just supposed to indicate direction of travel. If the Ice divide label and arrow were spaced further apart, it might be more obvious that the ice divide is the path from 2-3. And maybe make the inset a little larger and label the ice divide on the map of Greenland? Also, just a suggestion if it doesn't make it too cluttered is to add more labels (i.e., the 60.7 degrees from north) Figure 6: a-d should be described in the caption, i.e., the caption should say something like, "Averaged power-depth profiles along the straight line perpendicular to the divide for the 4 different polarization combinations (a-d). Figure 14: hard to read all the axis labels due to the resizing of the text, especially the x-axis title and labels, and the vertical labels as well. Figure 16: see note about Page 19...should define the thin section plane, or rename it in terms of the ice axis. Also, if the mention of "thin section" is left in this caption, the methods used to determine COF should be better described on page 19. At the very least, thin sections should be mentioned. Figure 17: The text of the figure labels looks strange due to resizing the figure (as do the data markers). Everything looks stretched out.

---

## Referee Comment (RC3) · H. Conway (Referee) · 28 Jun 2018

Dear Jilu, I appreciate your clarifications to my questions/comments. thank you, Howard

—————————————————————

---

## Author Comment (AC2) · 28 Jun 2018

Dear Prof. Courville,

Thank you very much for reviewing the manuscript and for your valuable comments to improve our manuscript. We would like to first address your specific and technical comments/questions (in red), and then integrate them into the final manuscript.

Specific Comments:

Page 5, Line 6: need more technical information about the GPS receiver.

We revised the sentence of line 6 on page 5 and added more information about the GPS receiver as follows (in green):

> We also installed a Topcon Legacy E GPS receiver between the transmitters and receivers to record geolocations with NMEA $GPGGA messages every 0.1 seconds.

Page 9, line 11: The truss, since it has such a large impact on the transmit power mismatch, should be introduced and explained in more detail (geometry? Construction? Materials?) in Section 2.

We revised line 11 on page 9 as below:

> The 12.7-dB transmit power mismatch between TX1 and TX2 is because of the aluminum truss effect …

We revised line 17 and 18 on page 9 as below:

> The real truss structure is much more complicated than the simulated one (see Fig. Sxx),…

We will provide the following picture in supplementary document to illustrate truss construction and geometry:

[Figure]

Fig. xx A picture of the truss construction and geometry

Page 19, section 4: suggest adding a brief description of the COF methods, for instance, in Figure 16 there is mention of thin sections. It would probably be good to mention that the COF are obtained from thin sections in the text. (or in the caption of Figure 16, instead of saying "projected into the thin section plane" name what that plane (guessing that it is vertical, but actually not sure).

We did not give a brief description of the COF methods because the cited references describe them in detail. We revised the first sentence in Section 4 as below:

> The COF along the full NEEM ice core was measured from vertical thin sections of the core in the field by an automatic ice texture analyzer every 10 m from a depth of 33 m down to 2461 m [Weikusat and Kipfstuhl, 2010], presented using the second-order orientation tensor $\boldsymbol{a}^{(2)}$ [Eichler et al., 2013; Montagnat, et al et al., 2014].

We revised the caption of Fig. 16 as below:

> Figure 16: Fabric represented in stereographic projection/pole figure projected into the **vertical planes of the thin sections** at depths of 330, 561, 849.8 and 1025.8 m. The *OY* direction in the pole figure is approximately the vertical ice core axis. The center of the diagram corresponds to the approximate horizontal. Only 10,000 data points randomly selected are plotted for each thin section. The color bar corresponds to the density of plotted data points.

Page 23, line 7: Earlier (Page 9, line 13), there is mention that the power reduction is due to these two factors as well as other unknown factors. That there are unknown factors contributing to the power reduction should also be mentioned in the summary.

We revised the summary as below:

> …, we identified a power reduction of about 12.7 dB transmitted towards the nadir in V-polarization because of the effects of the structure truss, mutual coupling on the antenna radiation pattern, and other unknown reasons. …

Technical Comments:

Throughout: don't need a comma after the name and before "et al." in citations (i.e., in lines 10 and 11), but do need a comma after et al. (i.e., line 17).

We have removed the comma after the name and before "et al."  and added a comma after et al. in all citations. Thanks for pointing out these minor but important things.

Page 1, Line 19: define VHF.

We have revised it to "VHF (very high frequency)".

Page 3, Line 15: Why is km bold and italicized?

We have corrected it.

Page 3, Line 24: should be, "plane also rotates." (or "the planes also rotate..") if this is referring to the vertical planes.

We have revised it to "the plane also rotates". Thanks for pointing out this grammatical error.

Page 4, Line 7: should be comma after "Marathe" in citation.

We have added a comma after "Marathe". Thanks for pointing out this.

Page 5, line 4: "snow tractor" is not a term I've heard before. We typically call these "tracked vehicles."

We have revised "snow tractor" to "tracked vehicle". Thanks.

Page 5 and throughout: should be consistent if there is a space after period in Fig or not (i.e., there is a space on page 4, Line 25 after Fig. 3, but not on lines 4 and 5 on page 5.

We have added a space after "Fig." on line 4 and 5 on page 5, and kept this consistent throughout the manuscript. Thanks.

Page 6, line 22: why is the m in bold?

We have corrected it.

Page 8, several instances: why is dB in parentheses here, and not above? Should be consistent at least.

We ignored the dB units in the equations, so we put dB in parentheses at the end of these equations.

Page 9, line 9: Fig. 7 should be capitalized.

We have corrected this. Thanks.

Page 11, line 4: seems like you should say something along the lines of, "The plane wave is of linear, circular, or elliptical polarization when this curve is, respectively, a straight line, a circle, or an ellipse, [Ulaby, 1981]." To make this sentence clearer.

We revised the sentence as suggested as below:

> The plane wave is of linear, circular, or elliptical polarization when this curve is, respectively, a straight line, a circle, or an ellipse [Ulaby, 1981].

Page 17, line 15: do the authors mean, optic axis instead of optical axis here?

We have replaced "optical" with "optic" throughout the manuscript. Thanks for pointing out this.

Page 19, line 21: think that this is actually supposed to be referring to Figure 15, not Figure 14.

We have corrected this. Thanks.

The variable beta was defined at line 20, page 12. We reiterate it here by adding the following:

"where $\beta$ is the angle between the principal axis $z_p$ and the z-axis,…".

Page 22, line 21: This is the first mention of thin sections in the text. I think there should be a brief discussion of the methods on Page 16, or refer to these as "samples," but be consistent.

Please see our responses to your specific comments about Page 19.

We have corrected this typo. Thanks for pointing out it.

Figure 2: Maybe add a label with the dipole lengths in the figure?

We have labeled the dipole lengths as suggested, see the revised Fig. 2 below:

[Figure]

Figure 5: Better label the ice divide? Right now, it looks like the "Ice divide" label is associated with the arrow, which I think is just supposed to indicate direction of travel. If the Ice divide label and arrow were spaced further apart, it might be more obvious that the ice divide is the path from 2-3. And maybe make the inset a little larger and label the ice divide on the map of Greenland?

Also, just a suggestion if it doesn't make it too cluttered is to add more labels (i.e., the 60.7 degrees from north)

We have revised Fig. 5 and its caption to avoid the ambiguity, as shown below:

[Figure]

(a)                                        (b)

**Figure 5: Paths of multi-polarization measurements: (a) Paths along ice divide and perpendicular to ice divide. (b) Circular paths at turns. The paths are plotted in the local ENU (East-North-Up) coordinate system at the NEEM ice core camp site (77.45°N 51.06°W), marked by a red dot at (0,0). The travel direction is illustrated by the arrows along the paths. The local ice divide is along the path from location 2 to 3, as marked by the dashed line with arrows on both ends. The red dot and the dashed line with arrows in the inset map show the location of the NEEM site and the direction of the local ice divide in Greenland. The cross formed by a red and a blue segment shows the location of a crossover where the data were collected with the non-polarimetric HH polarization antenna configuration (see discussions on Fig. 14).**

Figure 6: a-d should be described in the caption, i.e., the caption should say something like, "Averaged power-depth profiles along the straight line perpendicular to the divide for the 4 different polarization combinations (a-d).

We have revised the caption of Fig. 6 as below:

> **Figure 6: Averaged power-depth profiles along straight line perpendicular to the ice divide (from pt. 1 to pt. 4 in Fig. 5) for the four different Tx-Rx polarization combinations. (a) HH and VH; (b) HV and VV; (c) HH and HV; and (d) VH and VV. The insets show three specular reflection peaks from layers at depths of 1142, 1181 and 1271 m.**

Figure 14: hard to read all the axis labels due to the resizing of the text, especially the x-axis title and labels, and the vertical labels as well.

We will pay attention to and address the above resizing issues during the final submission of this manuscript.

Figure 16: see note about Page 19…should define the thin section plane, or rename it in terms of the ice axis. Also, if the mention of "thin section" is left in this caption, the methods used to determine COF should be better described on page 19. At the very least, thin sections should be mentioned.

Please see our responses to your specific comments about Page 19.

Figure 17: The text of the figure labels looks strange due to resizing the figure (as do the data markers). Everything looks stretched out.

We will pay attention to and address the above resizing issues during the final submission of this manuscript.

---

## Author Response (AR1)

Dear Editor,

We have uploaded a revised version of the manuscript based on the comments from referees. We would like to first thank all of the referees for their valuable comments, which have helped with improving the manuscripts.

In the revised version and the supplementary sections, all changes are marked in red font. Below we list our responses (in black) item by item to each of the referee's comments and questions (in blue):

Referee 1 (Howard Conway):

Others have shown that depth-profiles of radar-detected fabric show good agreement with direct measurements of fabric from ice cores (e.g. Drews et al. 2012, Eisen et al., 2007 at EDML; Fujita et al, 2006 at Dome Fuji and Mizuho). Here, the authors report multi-channel and multi-polarization radar measurements in the vicinity of the NEEM ice-core site. The radar measurements are used to infer depth-profiles of birefringence and fabric and compared with published measurements along the core (Montagnat et al. 2014 and Weikusat et al. 2017).

The new measurements, together with those from previous work, gives confidence that profiles of ice fabric in ice sheets can be determined using remote sensing methods. Given the importance of ice-fabric on ice dynamics, and the logistical difficulties of extracting and processing ice cores, this is an important contribution.

Thanks for the very positive comments.

A few comments/questions: Sections 2, 3.1 & 3.2 provide in-depth descriptions and details of a very impressive radar system, data collection and processing methods. You also note a power mismatch between the two transmitters when used in HH and VV orientations, and provide a simulation to estimate the influence of the truss on the antenna radiation pattern. Conclusion is that the power mismatch was likely caused by interference between the radiation patterns. The focus for the remaining part of the paper is on polarization measurements from just one of the transmitters, which includes HH and HV transmit-receive polarizations. In order to make the manuscript more accessible to readers who are not so interested in details of the radar system, I suggest you consider moving these detailed descriptions (together with Figs. 1, 2, 3, 4, 6, 7 & 8) into supplementary information.

As suggested, we moved the detailed description of the radar system into supplementary section S1 and Fig. 1 into Fig. S1. We moved the description of the antenna geometry, radiation patterns and field installation into supplementary section S2, together with Figs. 2, 4 and 8 into Figs. S2, S3 and S4. We would like to keep Fig. 3 as now Fig. 1 in the paper because we think it is essential to understanding the rest of the paper. We moved the detailed discussions about the receive channel equalization into supplementary section S3, together with Fig. 6 into Fig. S6. We moved the detailed discussions on the transmit power mismatch between H and V orientations into supplementary section S4, together with Fig. 7 into Fig. S7.

In order to keep the manuscript accessible to readers, you might also consider moving the discussion of filtering data (together with Figs 11&12) to the supplementary section.

As suggested, we moved the discussion of data filtering into supplementary section S5, together with Fig. 12 into Fig. S8. We would like to keep Fig. 11 as now Fig. 5 in the paper because it is the basis for the later analyses in the rest of the paper.

Detailed questions

Please see https://www.the-cryosphere-discuss.net/tc-2018-94/tc-2018-94-AC1-supplement.pdf uploaded on June19, 2018. Also see the revised manuscript and the supplementary sections.

Referee 2 (Zoe Courville):

General Comments: This is a general well-written manuscript that presents very interesting results comparing multi-channel/multi-polarization radar results to measured crystal orientation fabric from the NEEM ice core. The results show good comparison between the two methods. The methods are well-presented and clear.

Thanks for the very positive comments.

Specific Comments

Please see https://www.the-cryosphere-discuss.net/tc-2018-94/tc-2018-94-AC2-supplement.pdf uploaded on June 28, 2018. Also see the revised manuscript and the supplementary sections.

In addition to the changes based on the referees' comments, we further improved the manuscript by including an anisotropic simulation to support the analysis and discussion on the observed anisotropic patterns in HH measurements. Please see this improvement by the added Fig. 4b and modified Fig. 4c, and the added discussion at the end of the first paragraph on page 9 in the revised manuscript.

Sincerely

Jilu Li, corresponding author